# Plasma cell differentiation is regulated by the expression of histone variant H3.3

Yuichi Saito[1,2], Akihito Harada[3], Miho Ushijima[1], Kaori Tanaka[3], Ryota Higuchi [1,2], Akemi Baba[1], Daisuke Murakami [2], Stephen L. Nutt [4,5], Takashi Nakagawa [2], Yasuyuki Ohkawa [3] ✉ & Yoshihiro Baba [1] ✉

The differentiation of B cells into plasma cells is associated with substantial transcriptional and epigenetic remodeling. H3.3 histone variant marks active chromatin via replication-independent nucleosome assembly. However, its role in plasma cell development remains elusive. Herein, we show that during plasma cell differentiation, H3.3 is downregulated, and the deposition of H3.3 and chromatin accessibility are dynamically changed. Blockade of H3.3 downregulation by enforced H3.3 expression impairs plasma cell differentiation in an H3.3-specific sequence-dependent manner. Mechanistically, enforced H3.3 expression inhibits the upregulation of plasma cell-associated genes such as *Irf4, Prdm1*, and *Xbp1* and maintains the expression of B cell-associated genes, *Pax5, Bach2*, and *Bcl6*. Concomitantly, sustained H3.3 expression prevents the structure of chromatin accessibility characteristic for plasma cells. Our findings suggest that appropriate H3.3 expression and deposition control plasma cell differentiation.

Through the secretion of antibodies, plasma cells play an important role in humoral immunity and long-term host protection against infections. When activated by antigens or Toll-like receptor (TLR) ligands, B cells differentiate into short-lived antibody-secreting plasmablasts and further develop into long-lived plasma cells (both of which are referred to as plasmacytes [PCs] herein)[1,2]. Although antibodies are essential for humoral immunity, pathogenic antibodies are often key mediators of autoimmune diseases[3]. PCs also modulate immune responses through the secretion of inflammatory or anti-inflammatory cytokines which are related to the pathophysiology of various diseases[4,5].

During terminal differentiation into PCs, the loss of B cell identity occurs, and the transition from B cells to PCs is associated with extensive transcriptional and epigenetic rewiring[6,7]. Changes in gene expression involve the upregulation of genes specific to the PC program (e.g., *Irf4, Prdm1*, and *Xbp1*)[8–12] and the repression of genes specific to the B-cell program (e.g., *Pax5, Bach2, Bcl6, Spi1*, and *Irf8*)[2,13–19]. Lack of *Pax5, Bach2, Bcl6*, or *Spi1* in B cells facilitates differentiation into PCs, as these factors appear to inhibit PC transition[16,18–21]. PC differentiation is initiated by IRF4 via the activation *of Prdm1* (encoding Blimp1)[8,9]. In B-lineage cells, Blimp1 is only expressed in PCs and is essential for cell differentiation[10,22]. Blimp1 silences B cell lineage transcription factors such as Bcl6 and Pax5 which concurrently have the potential to inhibit *Prdm1* expression and PC differentiation[18,19,23–25]. Xbp1 coordinates diverse changes in cell structure and function, resulting in the maturation of antibody-secreting PCs[26]. Epigenetic modifications such as DNA methylation and histone post-translational modifications are dynamically altered at different stages of B cell differentiation. In mice and humans, DNA methylation is predominantly lost when B cells differentiate into PCs. The unique DNA methylation landscapes of PCs have been defined, and their dynamic changes, along with transcriptional alterations, have been observed during cellular division-dependent PC differentiation[27,28]. The loss of ten-eleven translocation (TET) proteins, enzymes that facilitate DNA demethylation and recruit histone deacetylases, leads to impaired germinal center (GC) B cell and PC differentiation[29] and a break in

[1]Division of Immunology and Genome Biology, Medical Institute of Bioregulation, Kyushu University, Fukuoka, Japan. [2]Department of Otorhinolaryngology, Graduate School of Medical Sciences, Kyushu University, Fukuoka, Japan. [3]Division of Transcriptomics, Medical Institute of Bioregulation, Kyushu University, Fukuoka, Japan. [4]The Walter and Eliza Hall Institute of Medical Research, Parkville, VIC 3050, Australia. [5]Department of Medical Biology, The University of Melbourne, Parkville, VIC 3010, Australia. ✉e-mail: yohkawa@bioreg.kyushu-u.ac.jp; babay@bioreg.kyushu-u.ac.jp

peripheral tolerance[30]. Histone modifications are critically involved in changes in gene expression during B cell differentiation and are important for PC differentiation[7,27]. In addition to epigenetic DNA methylation and histone modifications, histone variants also regulate the expression of selected genes during cell development[31–34]. Although the transcription factors and genes they regulate have been studied epigenetically, little is known about the roles of histone variants in PC differentiation.

Most mammals have two similar histone H3 family members, canonical H3 and non-canonical H3.3 variant. H3.3 is encoded by *H3f3a* and *H3f3b* whose transcription translates to an identical protein[35] and differs from the canonical histone H3.1 by just five amino acid substitutions, which are important for the specific and crucial regulation of chromatin dynamics and transcription[36,37]. Current evidence indicates that H3.3 is a key player in diverse processes, including regulation of mammalian ontogeny and cell differentiation[38–40]. The replication-independent deposition of H3.3 occurs in many transcriptionally active regions and is closely associated with active histone marks[41,42]. Furthermore, hematopoietic stem cells (HSCs) lacking H3.3 do not give rise to B cells[43] suggesting that H3.3 is important for the early development of B cells. However, the dynamics of H3.3 and its role in PC differentiation remain unresolved.

In the present study, we aim to determine the impact of H3.3 on the differentiation of B cells into PCs and investigate the mechanisms involved in regulating this process. We find that H3.3 transcription and nucleosome deposition were downregulated during PC differentiation. PCs have distinct chromatin accessibility compared to B cells, and this difference is associated with H3.3 distribution. Enforced expression of H3.3 prevents the downregulation of H3.3 and inhibits the transition of B cells into PCs in vitro and in vivo. Although H3.3 expression does not affect B cell proliferation and apoptosis, it suppresses the expression of *Irf4*, *Prdm1*, and *Xbp1* and downregulation of *Pax5*, *Bcl6*, and *Bach2*. In addition, B cells expressing H3.3 are associated with decreased accessibility of PC-related genes, even after PC differentiation has been induced. Together, our data suggest an essential role for H3.3 in the dynamic changes in transcription governing B cell differentiation into PCs.

## Results

### H3.3 expression was reduced during plasma cell differentiation

To examine the role of H3.3 in PC differentiation, we first analyzed its expression level in both B cells and PCs. Quantitative PCR (qPCR) analysis of sorted naive splenic B cells and PCs isolated from the spleen and bone marrow of heterozygous *Prdm1*^gfp/+^ mice revealed that the levels of expression of *H3f3a* and *H3f3b* were markedly lower in PCs than in B cells (Fig. 1a). The level of expression of *H3c1/2*, which encodes canonical H3, was not different between B cells and PCs in the spleen; however, it was decreased in PCs in the bone marrow (Fig. 1a). Similar results were obtained using in vitro analyses. Splenic B cells from wild-type mice were stimulated with lipopolysaccharide (LPS) to induce B cell differentiation into PCs. We found that the mRNA expressions of *H3f3a* and *H3f3b*, but not of *H3c1/2*, were downregulated in PCs (Fig. 1b). Consistent with the above findings, the incorporation of H3.3 protein into the genome was also reduced in PCs (Fig. 1c). The effect of cell division on the expression of H3.3 was further investigated because LPS-induced PC differentiation requires cell division. Four days after being treated with LPS, *Prdm1*^gfp/+^ B cells, undivided B cells (CD138⁻ Blimp1-GFP⁻), PCs (CD138⁺ Blimp1-GFP⁺), and the same number of dividing B cells (CD138⁻ Blimp1-GFP⁻) were sorted and assessed with qPCR. *H3f3a* and *H3f3b* expression decreased with cell division and declined remarkably in PCs (Fig. 1d). The level of H3.3 protein was also reduced in PCs (Fig. 1e). These results suggest that H3.3 was downregulated in accordance with the PC differentiation process.

### Dynamic deposition of H3.3 and chromatin structural change during PC differentiation

To detect dynamic changes in H3.3 localization during PC differentiation, we performed chromatin integration labeling (ChIL)-sequence (ChIL-seq) for H3.3 on naive B cells, activated B cells, and PCs after LPS stimulation[44]. As expected, high and consistent levels of H3.3 were observed in housekeeping genes during PC differentiation (Supplementary Fig. 1a), suggesting that our prepared H3.3 ChIL-Seq library is reliable for further downstream analysis. We found that H3.3 enrichment decreased during PC differentiation, which was consistent with the downregulation of H3.3 (Fig. 2a). Furthermore, the principal component analysis (PCA) of the H3.3 binding profile revealed the dynamic process of PC differentiation (Fig. 2b). Although H3.3 expression was reduced in PCs, chi-square tests revealed that the remaining H3.3 peaks localized at promoter regions were significantly increased in PCs (Fig. 2c and Supplementary Table 1). We used meme-chip to overlap H3.3 peak with known transcription-factor-binding motifs and observed that in activated B cells, H3.3 enrichment region "preferentially" occurred near motifs for Bach2 (*E*-value: 4.6e−006), and in PCs, near motifs for Prdm1 (*E*-value: 2.6e−008) whereas in naive B cells, the enrichment regions were extensive (Fig. 2d). Notably, ChIL-seq analysis revealed that on *Xbp1* and *J chain*, a dynamically increased enrichment of H3.3 occurred, while on *Irf4* and *Prdm1*, which are highly expressed in PCs, the expression of H3.3 was maintained (Fig. 2e). Meanwhile, reduced enrichment of H3.3 was observed on *Pax5*, *Bach2*, and *Bcl6*, which maintain B-cell identity. H3.3 deposition was not largely altered at *Spi1* and *Irf8* loci. These data suggest that during the differentiation of naive B cells into PCs, the deposition of H3.3 may be correlated with gene expression at each stage of differentiation.

Next, we performed transposase-accessible chromatin sequencing (ATAC-Seq) to explore chromatin structural changes that occur during PC differentiation. At the chromatin level, dynamic changes were observed in the accessibility profile of B cells during their differentiation into PCs. The PCA of the accessible loci detected in B cells and PCs suggested that progressive changes in accessibility occurred during PC differentiation (Fig. 2f), as previously reported[27]. The ATAC-seq distribution pattern was similar between naive B cells and PCs (Fig. 2g). Regions that are differentially accessible (DAR) were identified by comparing PCs cells to naive B cells. While overall, there were differences in the accessible region, with a similar number of regions gaining or losing accessibility, the enrichments of H3.3 were markedly higher in naive B cells than in PCs (Fig. 2h).

Next, we analyzed the genome-wide relationship between H3.3 uptake and accessibility during PC differentiation. Accessible regions that are specifically closed during PC differentiation (open-naive) exhibited a lower H3.3 peak count in PCs than in naive B cells (Fig. 2i). Accessible regions that are specifically opened during PC differentiation (open PC) presented an unaltered H3.3 read count (Fig. 2j). Enrichment of H3.3 on each DAR was annotated to the closest gene, and Gene Ontology (GO) analysis revealed that in PCs, genes enriched in H3.3 were correlated with DAR and showed high enrichment of biological processes, including B cell maturation (Fig. 2k). Taken together, these observations suggest a correlation between the patterns of H3.3 deposition, gene expression, and chromatin accessibility during PC differentiation.

### Enforced H3.3 expression in B cells suppresses PC differentiation

These findings led us to consider whether the differentiation of B cells into PCs is affected by H3.3 expression. To investigate the role of H3.3 in PC differentiation, we induced green fluorescent protein (GFP)-tagged H3.3 (GFP-H3.3) expression in B cells to compete with H3.3 in chromatin regions with high H3 turnover. GFP/yellow fluorescent protein-tagged H3.3 has been extensively used to study H3.3 replacement in Drosophila, mouse, and human cells[45–48]. GFP-H3.1 was also

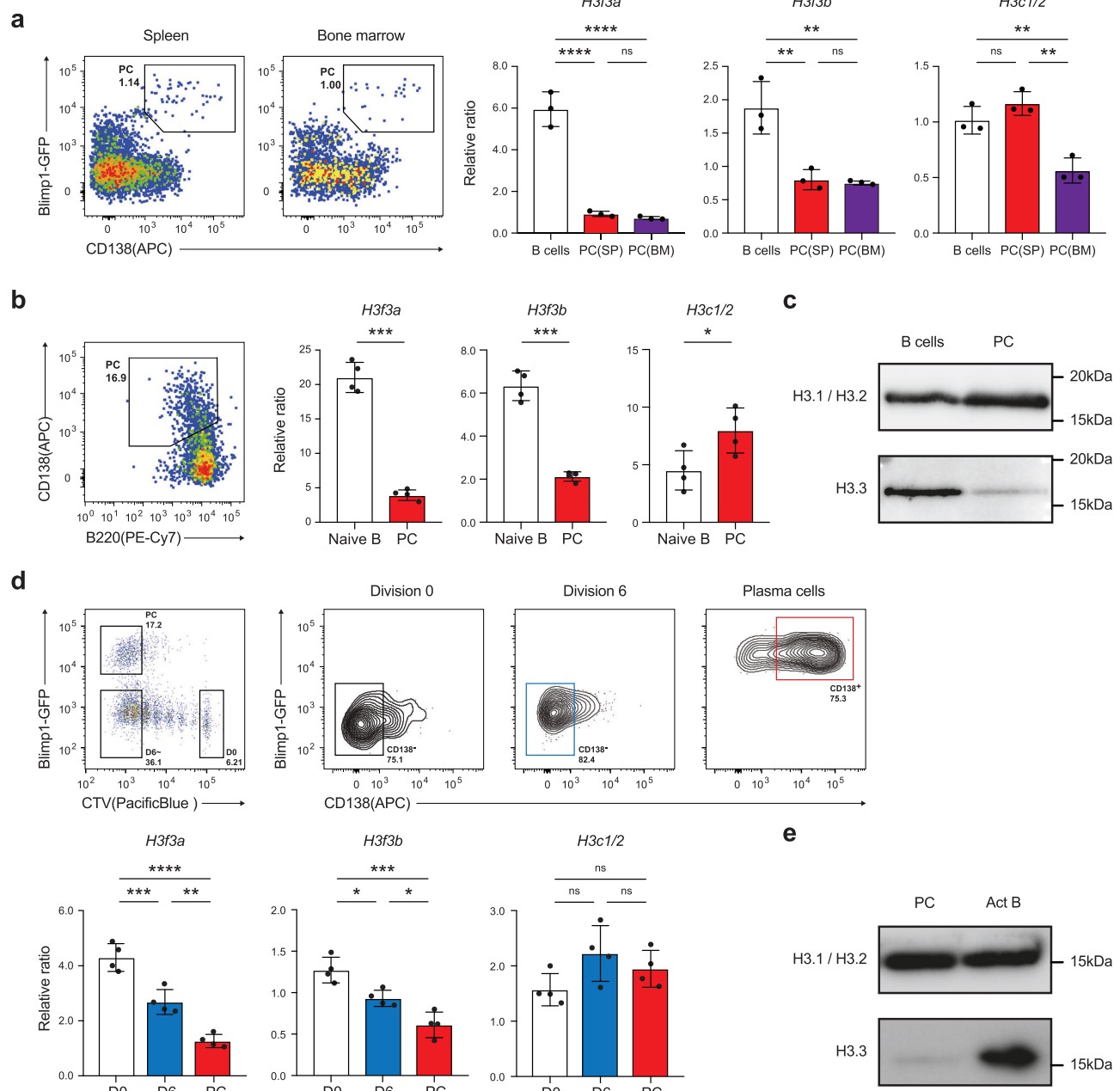

**Fig. 1 | H3.3 expression was reduced during PC differentiation. a** Quantitative RT-PCR of mRNA encoding H3.1/H3.2 (*H3c1/H3c2*) and H3.3 (*H3f3a* and *H3f3b*). B cells in spleen and CD138hiGFP+ plasma cells (PC) in spleen or bone marrow were isolated from naive *Prdm1*gfp/+ mice. On the left, the frequency of plasma cells is shown. **b** Quantitative RT-PCR of mRNA encoding H3.1/H3.2 and H3.3 in naive splenic B cells or CD138hiB220low plasma cells cultured with LPS for 4 days. **c** Western blotting analysis of Hydroxyapatite (HAP)-purified chromatin from naive splenic B cells or LPS-induced plasma cells (PC) with antibodies specific for H3.1/H3.2 and H3.3. **d** Representative flow cytometry plots to analyze the cell-cycle status of CTV-labeled B cells from *Prdm1*gfp/+ mice on day 4 after LPS stimulation. Quantitative RT-PCR of mRNA encoding H3.1 and H3.3 in B cells at each stage of cell division. 0 (D0) and over 6 times division fraction among GFP−CD138− activated B cells (D6) and GFP+ CD138+ plasma cells (PC) among over 6 times division fraction were sorted. Examples of cells were pre-gated on live (PI−) single lymphocytes (**a**, **b**, **d**). **e** Western blotting analysis of LPS-induced PC and activated B cells with antibodies specific for H3.1/H3.2 and H3.3. Data are pooled from two (*n* = 3) (**a**) or three (*n* = 4) (**d**) independent experiments or representative of two independent experiments (**b**; *n* = 4, **c**, **e**; *n* = 2). Data are presented as mean ± SD. *$p < 0.05$; **$p < 0.01$; ***$p < 0.001$; ****$p < 0.0001$; ns not significant. The *p* values were obtained by one-way ANOVA with Tukey's post hoc test (**a**, **d**) or a two-tailed unpaired *t*-test with Welch's correction (**b**). Source data and exact *p* values (**a**, **b**, **d**) are provided as a Source Data file.

transduced to compare its function to that of H3.3. Fluorescence microscopy confirmed the distribution of GFP-H3.1 in the condensed chromosomal region (heterochromatin) and that of GFP-H3.3 in the non-condensed chromosomal region (euchromatin) (Fig. 3a). This is consistent with the enrichment pattern of endogenous H3.1 and H3.3[35,48]. After transduction of the control and GFP-H3.3/H3.1, B cells were induced to differentiate into PC by LPS stimulation. We found

that exogenous H3.3 expression significantly decreased the frequency of PC generation, while control and H3.1 expression did not alter it (Fig. 3b). The total IgM concentration in the culture supernatant was measured using ELISA. The results were consistent with those of flow cytometry (Fig. 3c). Similar results were obtained for mCherry-tagged H3.3-expressing B cells (Supplementary Fig. 2a, b). The secretion of IL-10, which is also an important regulator of PC biology, was markedly

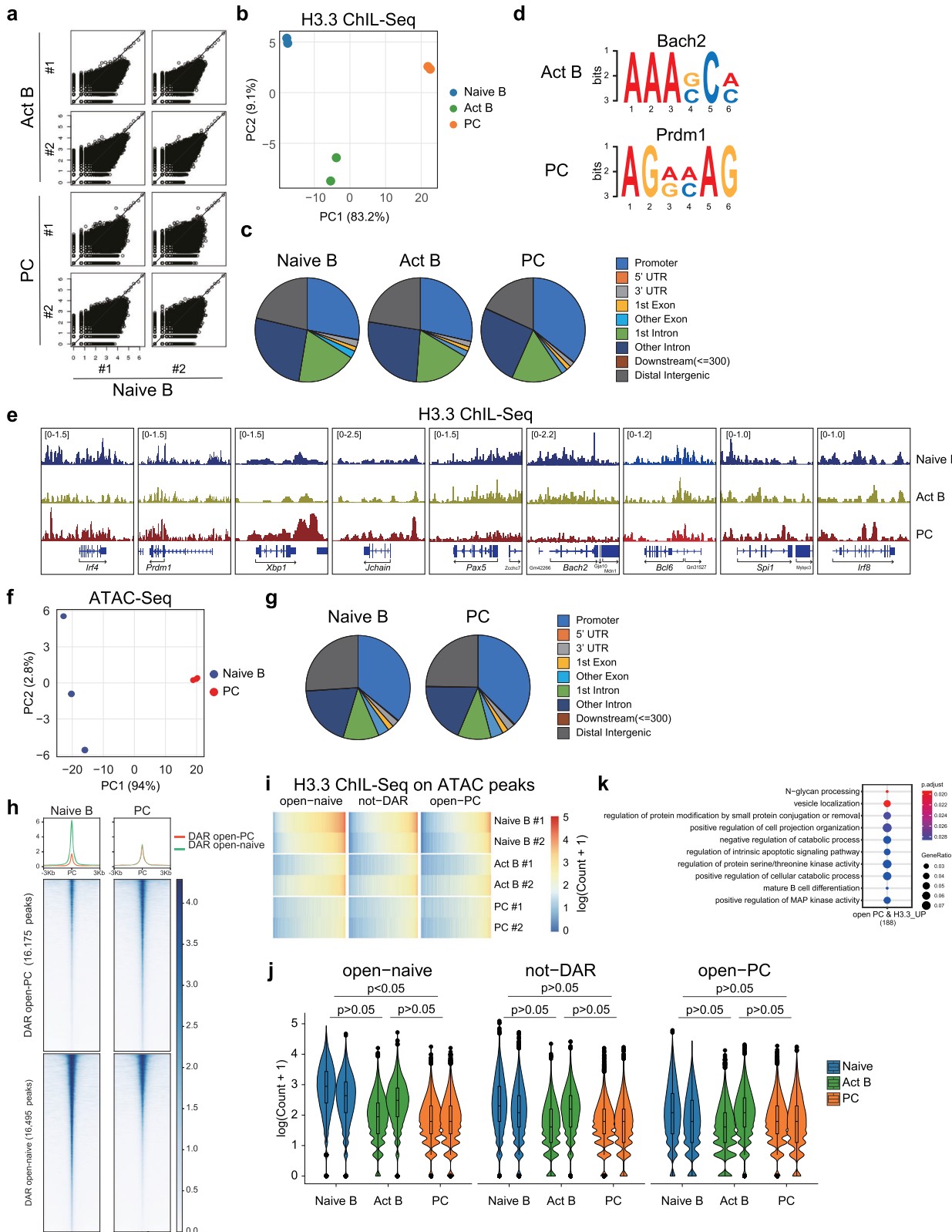

impaired by the enforced expression of H3.3 (Supplementary Fig. 2c). We also evaluated the class switch from IgM to IgG1, but forcibly expressed-H3.3 did not influence this event (Supplementary Fig. 2d). Given that LPS-induced PC differentiation depends on cell proliferation, we examined the effect of H3.3 expression on the latter. We labeled purified B cells with cell trace violet (CTV) before transduction with H3.3 and monitored cell division by CTV dilution assay. LPS

stimulation induced several rounds of B cell division, regardless of whether H3.3 was introduced (Fig. 3d). Furthermore, B cells transduced with H3.3 survived after LPS stimulation (Fig. 3e). These results indicated that the H3.3-induced inhibition of PC generation is not due to impaired proliferation or survival. Additionally, H3.3 expression also suppressed PC differentiation and antibody production in anti-CD40 and cytokine stimulation, a model of T cell-dependent PC

**Fig. 2 | Dynamic change of H3.3 deposition of H3.3 and chromatin accessibility during PC differentiation. a** Scatter plots of H3.3 ChIL-seq signals in 4 K bins for H3.3 enrichment. "#1" and "#2" represent two individual experiments. **b** Principal component analysis (PCA) of the H3.3 signal in 4K-bins between naive B (naive), activated B cells (Act B), and plasma cells (PC). **c** Genomic feature distributions of H3.3 ChIL-seq reads. Chi-square two-sided tests are shown (H3.3 ChIL-seq peak in PCs overlapping the promoter region: $p = 1.99214 \times 10^{-72}$ for naive, $p = 4.2276 \times 10^{-51}$ for activated B). **d** Enrichments of known motifs at Act B and PC occupied regions identified in H3.3 ChIL-seq peaks from both replicates. Transcription factors potentially recognizing the motifs are displayed. **e** Profiles of ChIL-seq data for H3.3 are shown in naive, act B, and PC. **f** PCA of ATAC-seq signals in 4K-bins between naive and PC. **g** Genomic feature distributions of ATAC-seq reads in (**c**). **h** Heatmap of ATAC-seq signals in the differential accessible regions (DARs) within ±3 kb around peak centers. DARs were classified into two categories (open-naive and open-PC) based on the presence of ATAC-seq peak between naive and PC. The values are average normalized ATAC-Seq signal intensity. **i** Heatmap of H3.3 ChIL-seq signals in the DARs or not-DARs within ±2 kb. **j** Violin plots of H3.3 enrichments on the DARs or not-DARs within ±2 kb. The lower and upper hinges represent the 25th and 75th percentiles, respectively. Whiskers extend up to 1.5 * IQR (interquartile range) from the hinges. The $p$ values are calculated from two-sided $t$-tests using lmerTest (R package) (biologically independent two experiments). **k** Gene ontology (GO) analysis for genes around the region of high accessibility and high H3.3 enrichment in PC. The number of genes in the analyzed indicates under the label. The bubble size indicates the gene enrichment ratio (GeneRatio) of a biological process GO term, with color maps of the FDR value (p.adjust) of the enrichment analysis, log2 FC < 1. $p < 0.05$; a one-sided Fisher's exact $t$-test. Source data and exact $p$ values (**j**) are provided as a Source Data file.

differentiation (Supplementary Fig. 2e, f). Conversely, H3.3 knockdown enhanced PC differentiation following stimulation with LPS or CD40 and cytokines (Supplementary Fig. 2g–j).

To assess whether H3.3 expression was sufficient to limit PC differentiation in vivo, we transduced GFP-H3.3 or control vector into B cells and then transferred them into B cell-deficient μMT recipient mice 1 day before immunization with LPS (Fig. 3f). Transduction with H3.3 significantly attenuated the frequency of PCs compared with that of the control vector. Thus, these data strongly suggest that enforced H3.3 expression in B cells suppresses PC differentiation.

### AIG motif is required for H3.3-induced inhibition of PC differentiation

The amino acid sequence of H3.3 differs from that of H3.1 at two sites. S31 in H3.3 (corresponding to the location of A31 in H3.1) is phosphorylated during mitosis; however, the role of this modification is unknown. H3.3 also has three residues at the base of α-helix 2, which differ from H3.1. These are A87/I89/G90 (hereafter referred to as "AIG"), which correspond to S87/V89/M90 in H3.1. This motif is thought to define the chaperone specificity of H3.3. The AIG motif is required for H3.3 binding to both the death-associated protein (DAXX)/α-thalassemia X-linked mental retardation protein (ATRX)[49] and histone regulator A (HIRA) complexes[50], which are fundamental for replication-independent chromatin deposition. To assess the effects of individual residues, single mutations were introduced into H3.3 gene (with amino acid residues S31, A87, I89, and G90) to match each residue of interest in H3.1 (A31, S87, V89, and M90). These mutants were used in the LPS-mediated PC differentiation assays. Expression of GFP-H3.3 carrying the mutations A87S, G90M, and A87S/I89V/G90M, led to normal PC generation (Fig. 4a, b). The S31A mutation did not affect the inhibitory effect of H3.3 on PC differentiation. These findings suggest that the inhibition of PC differentiation by H3.3 may be required for AIG-dependent chaperones.

### Enforced H3.3 expression in B cells limits expression of Xbp1, IRF4, and Blimp1

PC differentiation is tightly controlled by several transcription factors that either promote or inhibit the expression of specific genes involved in PC differentiation. B cell-associated transcripts, such as those that encode Pax5, Bach2, IRF8, Bcl6, and PU.1, are suppressed to facilitate PC differentiation. Conversely, genes involved in PC differentiation, such as *Irf4*, *Prdm1*, and *Xbp1* (spliced form *Xbp1s*; unspliced form *Xbp1u*), are activated. We next examined whether enforced expression of H3.3 in B cells affected gene expression. qPCR revealed that transcripts of PC-related genes, such as *Prdm1* and *Xbp1s*, but not *Irf4* and *Xbp1u*, were significantly reduced by H3.3 expression (Fig. 5a). In contrast, the mRNA expression of B cell-related genes, such as *Pax5*, *Bach2*, and *Bcl6*, but not of *Spi1* and *Irf8*, was higher in H3.3-expressing B cells than in the control and H3.1-expressing B cells. Our data suggest

that H3.3 expression is associated with dynamic changes in the levels of expression of PC differentiation-associated genes.

IRF4 and Blimp1 are critical for the transition of B cells to PCs. To gain insight into the inhibitory effect of H3.3 on PC differentiation, the levels of expression of IRF4 and Blimp1 proteins were tested after H3.3 was expressed. We observed that enforced H3.3 expression, but not that of H3.1, reduced the number of IRF4 or Blimp1-positive cells (Fig. 5b, c). Notably, when IRF4 or Blimp1 were transduced into H3.3-expressing B cells, these cells efficiently differentiated into PCs (Fig. 5d), suggesting that IRF4 and Blimp1 can overcome the suppressive effect of H3.3. Thus, our findings suggest that forced expression of H3.3 before PC development preserves genes that maintain the B cell status and inhibits the expression of IRF4 and Blimp1, which are essential for PC differentiation.

### Enforced expression of H3.3 led to a dynamic change in chromatin accessibility profiles

To examine the effect of enforced H3.3 expression on chromatin structure, we first explored exogenous GFP-H3.3 deposition in GFP-positive LPS-stimulated B cells by ChIL-seq using an anti-GFP antibody. ChIL-seq following treatment of B cells with anti-H3.3 antibody was also performed to examine the total H3.3 (endogenous H3.3 and exogenous GFP-H3.3) deposition. GFP-ChIL-seq analysis of control vector-transfected B cells showed little signal, indicating negligible background (Fig. 6a). The deposition of GFP-H3.3 was similar to that of endogenous H3.3 in B cells transduced with an empty vector (Fig. 6a), suggesting that exogenous H3.3 integrate into chromatin and compete with endogenous H3.3. Furthermore, PCA revealed population-based clustering, and the differences increased with differentiation. Changes in H3.3 deposition in GFP-H3.3-expressing B cells were more similar to chromatin rearrangements observed in activated B cells than to those observed in PCs in both cases—exposure to anti-H3.3 and anti-GFP antibodies (Fig. 6b). To avoid the effects of retroviral expression and for direct comparison among cells, we used control vector-expressing activated B cells and PCs. These data suggest that PC differentiation requires proper H3.3 removal from B cells and specific H3.3 deposition. To determine the effect of H3.3 expression on chromatin accessibility, ATAC-seq was conducted in cells expressing GFP-H3.3 or the control vector. Given that the first principal component reflects PC differentiation, H3.3 expression led to a dynamic transition in chromatin accessibility profiles from those observed in PCs to those observed in activated B cells. This was not observed in the control vector-containing cells (Fig. 6c). Indeed, enforced H3.3 expression in B cells inhibits the accessibility of regions that increase with PC differentiation, such as the locus of *xbp1*, *prdm1*, *sdc1*, and *Jchain* (Fig. 6d). Thus, we conclude that the prevention of H3.3 downregulation during PC differentiation inhibits the changes in chromatin accessibility required for this process.

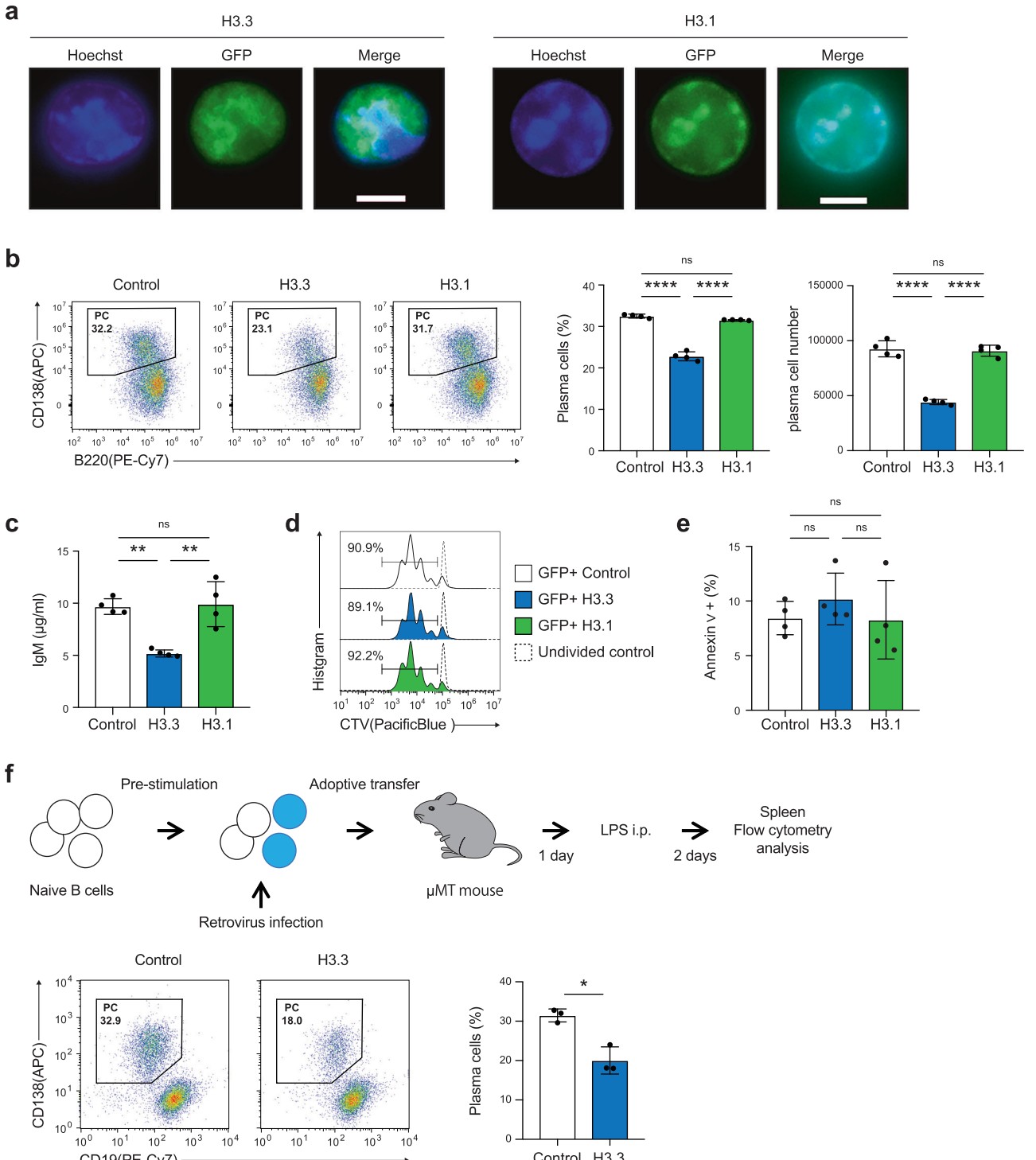

**Fig. 3 | Exogenous H3.3 expression inhibited PC differentiation. a** Fluorescence microscopic observation of GFP-H3.1 or GFP-H3.3-expressed B cells. B cells were stained with Hoechst. The scale bar is 5 µm. **b** Left, flow cytometry analysis of plasma cell differentiation in GFP (control), GFP-H3.3, or GFP-H3.1-expressing B cells on day 4 after LPS stimulation. The bars show the percentage (middle) or cell numbers (right) of PCs among GFP⁺ cells. **c** The IgM antibody concentration in the culture supernatant of (**b**) was measured by ELISA. **d** Proliferation of GFP⁺ B cells transduced with GFP (control), GFP-H3.3, or GFP-H3.1 labeled with CTV and stimulated for 3 days with LPS. Percentages of proliferating B cells were assessed by CTV dilution. **e** Apoptosis of GFP (control), GFP-H3.3, or GFP-H3.1-expressing B cells on day 4 after LPS stimulation, was evaluated by staining with Annexin V. **f** B cells were

transduced with GFP (control) or GFP-H3.3 were adoptively transferred to µMT mice and then immunized with 10 µg LPS. After 3 days, flow cytometry of spleen was conducted. Flow cytometry plot of plasma cell (PC) in GFP (control) or GFP-H3.3 expressing B cells. Examples of cells were pre-gated on live (PI⁻) single lymphocytes GFP⁺ cells (**a**, **b**, **d**, **f**). Data are representative of three (**a**, *n* = 1 mice per group) or two independent experiments (**b**–**f**) (**b**; *n* = 4, **c**; *n* = 4, **e**; *n* = 4, and **f**; *n* = 3 mice per group). Data are presented as mean ± SD. **p* < 0.05; ***p* < 0.01; ****p* < 0.001; *****p* < 0.0001; ns not significant. The *p* values were obtained by one-way ANOVA with Tukey's post hoc test (**b**, **c**, **e**) and by a two-tailed unpaired *t*-test with Welch's correction (**f**). Source data and exact *p* values (**b**, **c**, **f**) are provided as a Source Data file.

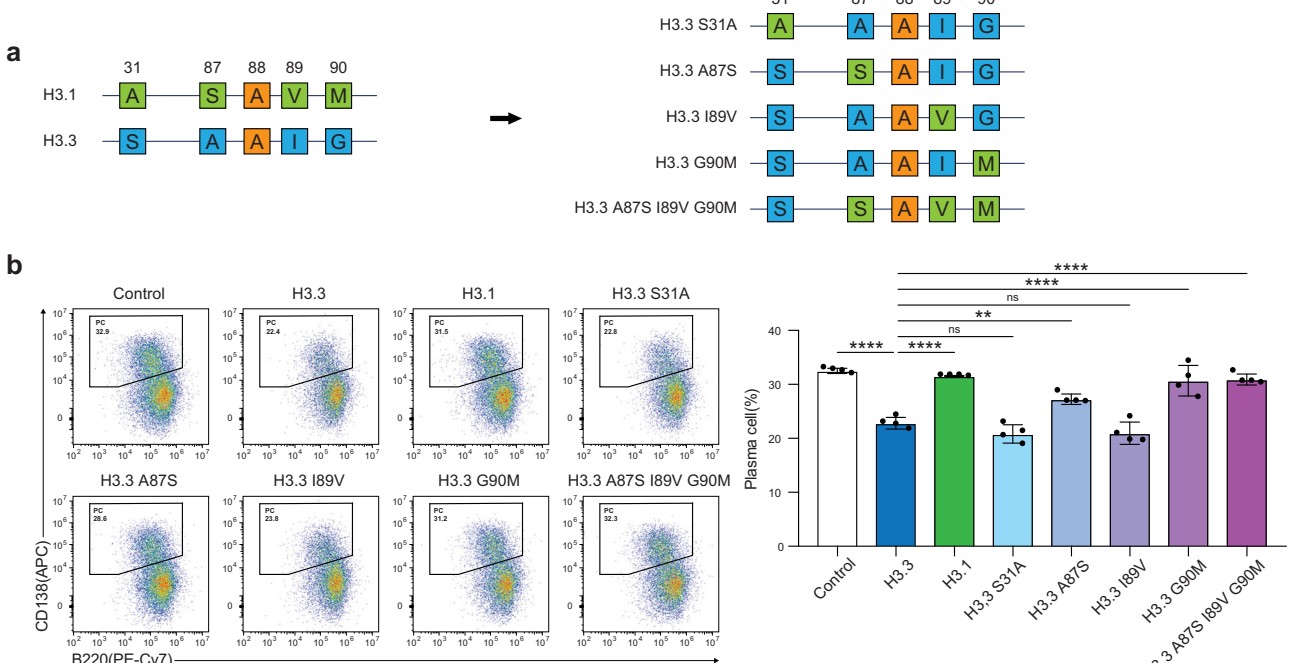

**Fig. 4 | A87 and G90 in H3.3 define suppressive function. a** Schematic diagram of the structure of H3.3 with point mutations in the N-terminal tail (S31A) and the "AIG" motif (A87S, I89V, and G90M). **b** Flow cytometry analysis of plasma cell differentiation. B cells were forcibly expressed H3.3 with the mutated residue to canonical H3.1 type via retrovirus infection and induced plasma cell differentiation with LPS. Examples of cells were pre-gated on live (PI⁻) single lymphocytes GFP⁺ cells. CD138hi B220low were gated as plasma cells (PC). Data are representative of two independent experiments. Data are presented as mean ± SD. *$p < 0.05$; **$p < 0.01$; ***$p < 0.001$; ****$p < 0.0001$; ns not significant. The $p$ values were obtained by one-way ANOVA with Tukey's post hoc test. Source data and exact $p$ values (**b**) are provided as a Source Data file.

## Discussion

Determining the molecular basis of B cell differentiation into PCs is important for understanding humoral immune responses. Our focus is on the role of the histone variant H3.3, a key regulator of the dynamic state of chromatin. Our findings indicate that H3.3 is downregulated during the differentiation of B cells into PCs. Accordingly, H3.3 incorporation into the genome and DNA accessibility are dynamically changed. Enforced expression of H3.3 in B cells results in a change in chromatin accessibility and suppression of IRF4, Blimp1, and Xbp1 expression, which leads to inhibition of PC generation.

Although only a few studies have examined the physiological significance of H3.3 expression levels, in mice, this variant seems to accumulate in various tissues and to almost completely replace the canonical isoforms with aging[51]. Surprisingly, in this study, we found that H3.3 expression and its genome-wide deposition in the DNA were remarkably reduced during PC differentiation. This might seem to contrast with the known roles of H3.3 in gene activation, a wide-spread process during PC differentiation. Given that H3.3 is not completely eliminated from PCs and that H3.3 deposition occurs on cis-regulatory elements of genes involved in PC differentiation, the dynamic change in the pattern of incorporation of H3.3 into chromatin may regulate the limited and appropriate gene expression required for PC differentiation. Indeed, we showed that when the PC differentiation program began, H3.3 was dissociated from the gene sites responsible for B cell identity, such as *Pax5*, *Bach2*, and *Bcl6*, and accumulated at the loci of PC-related genes, such as *Prdm1*, *Xpb1*, and *Jchain*. Although it is unclear whether the reduction of H3.3 protein levels is needed for PC differentiation, our results indicated that forced expression of H3.3 in differentiating B cells, halts PC generation, suggesting that the reduction of H3.3 protein levels is closely related to PC differentiation.

Recent evidence has demonstrated the importance of H3.3 in early B cell development. HSC-specific H3.3 conditional knockout mice exhibit completely impaired B-cell differentiation[43]. The deposition of H3.3 into chromatin is carried out by two different chaperones, HIRA and DAXX/ATRX[42,52]. HIRA incorporates H3.3 into the chromatin at the position of active transcription in a replication-independent manner[42,50]. DAXX/ATRX regulates H3.3 deposition in telomeric and pericentric heterochromatin regions[53]. Subsequently, the differentiation of HSCs from mice with conditionally knocked out HIRA or DAXX into B cells is impaired[54]. However, it is not known whether H3.3 and the specific chaperones are involved in PC differentiation. Our study showed that exogenously expressed H3.3 was incorporated into the genome to the same extent as physiological H3.3 and that inhibition of PC differentiation by H3.3 expression was dependent on the AIG motif of H3.3. In particular, the conversion of either A or G of AIG to the canonical form abolished the inhibitory effect of H3.3 on PC differentiation. Given that both A and G are required for HIRA binding and only one of them is sufficient for DAXX, the H3.3 inhibitory effect would be H3.3-specific chaperone-dependent, although we did not address the identification the specific chaperone. Additional experiments to understand the role of H3.3-chaperones for PC differentiation would also be of interest for future studies. Mechanistically, the abundant H3.3 expression in B cells, can limit the expression of IRF4 and Blimp1, and the activation of Xbp1, which results in the blockade of PC differentiation. Furthermore, even though the exogenous expression of H3.3 diminishes PC production, IRF4 and BLIMP1 overexpression with it reverses the suppression. Thus, the repression of gene expression required for PC differentiation by excess H3.3 provides a rationale for the reduction in PC generation among H3.3-overexpressing cells.

How does the enforced expression of H3.3 decrease PC-related gene expression? Chromatin accessibility is governed by multiple mechanisms, including DNA methylation, histone post-translational modifications, and transcription factor binding, which ultimately

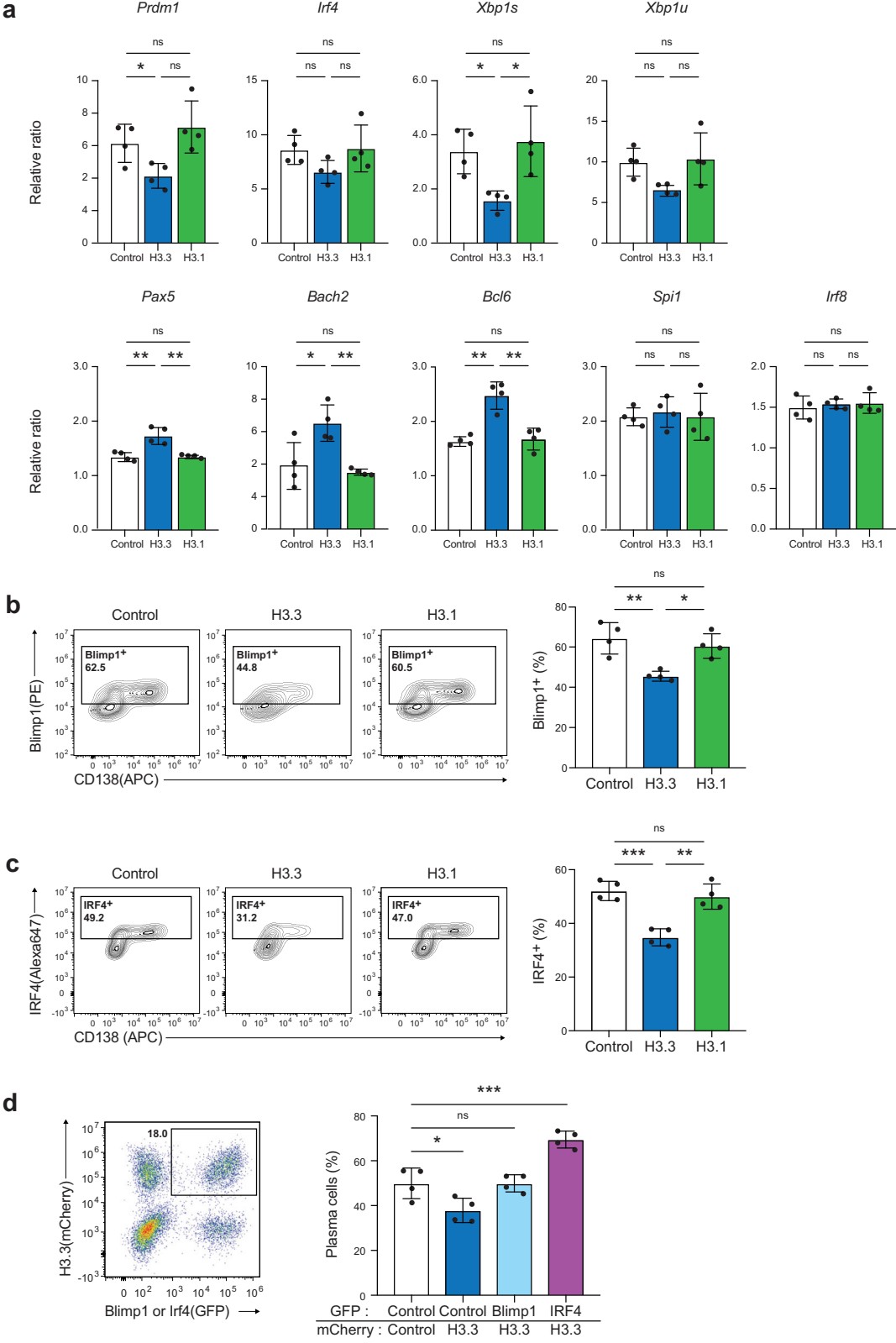

**Fig. 5 | H3.3 regulate the expression of PC-associated factors. a** Quantitative RT-PCR of mRNA encoding genes related to plasma cell differentiation in forcibly H3.1 or H3.3 expressed B cells. The GFP⁺ cells were sorted on post-infection day 2. **b** Flow cytometry analysis of the expression of the Blimp1 protein by intracellular staining in forcibly H3.1 and H3.3 expressed B cells. **c** Flow cytometry analysis of the expression of the IRF4 protein by intracellular staining in forcibly H3.1 and H3.3 expressed B cells. **d** Flow cytometry analysis of LPS-induced CD138^hi B220^low plasma cell differentiation after simultaneous enforced expression of H3.3 and Blimp1 or H3.3 and IRF4. Examples of cells were pre-gated on Zombie NIR⁻ (**b**, **c**) or Zombie Aqua⁻ (**d**) live cells. Data are representative of two independent experiments (**a**–**d**; $n = 4$). Data are presented as mean ± SD. *$p < 0.05$; **$p < 0.01$; ***$p < 0.001$; ****$p < 0.0001$; ns not significant. The $p$ values were obtained by one-way ANOVA with Tukey's post hoc test. Source data and exact $p$ values (**a**, **d**) are provided as a Source Data file.

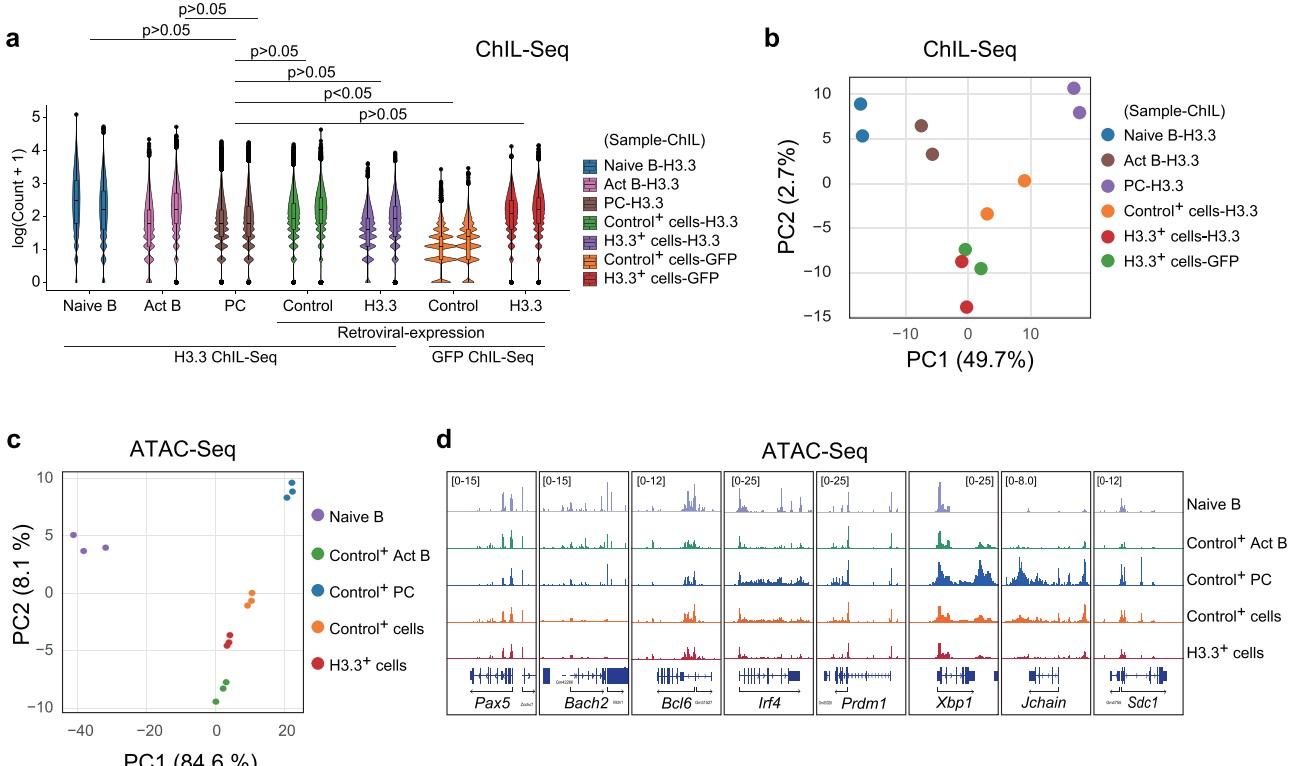

**Fig. 6 | Change of chromatin accessibility by enforced expression of H3.3.**
**a** Violin plots of H3.3 enrichments on chromatin accessibility regions within ±2 kb in naive B cells and plasma cells (PC). The lower and upper hinges represent the 25th and 75th percentiles, respectively. Whiskers extend up to 1.5 * IQR (inter-quartile range) from the hinges. The $p$ values are calculated from two-sided $t$-tests using lmerTest (R package) ($n = 65,822$, biologically independent two experiments). **b** PCA of H3.3-ChIL-seq in naive, activated B cells (Act B), PC, GFP-H3.3 expressed B cells and GFP-ChIL-seq in GFP-H3.3 expressed B cells. A contribution of the principal component is shown in parentheses. **c** PCA of ATAC-seq signals in 4K-bins in naive and forcibly GFP-H3.3 expressed B cells. A contribution of the principal component is shown in parentheses. **d** Genome browser images showing chromatin accessibility on the typical genes. Source data and exact $p$ values (**a**) are provided as a Source Data file.

regulate gene expression. Thus, it is important to disclose accessibility, such as the genomic state of regions that are prone to active transcription and those that are tightly bound to heterochromatin, to understand the relationship between chromatin structure and function. Several reports suggest that H3.3 may affect the chromatin environment in a specific area by recruiting particular complexes to the chromatin, including chromatin remodeling complexes and histone modification enzymes[33,35,55–57]. Thus, it controls gene expression and suppression. Our study provides even deeper insight: using ATAC-seq, we demonstrated that PC differentiation remodels the chromatin landscape, and H3.3 can contribute to the dynamics of this epigenetic landscape transition. Indeed, enforced expression of H3.3 altered accessibility and shifted it from that observed in PCs to that observed in activated B cells. It also reduced DNA accessibility at the gene loci of *Irf4*, *Prdm1*, and *Xbp1*, as well as genes specifically expressed in PC. These results suggest that retention of the H3.3 protein prevents changes in PC-specific accessibility, but this seems to contradict previous reports showing that H3.3 facilitates DNA accessibility[58]. One possible reason is that without lowering the protein levels of H3.3 in B cells, the PC differentiation program fails to be initiated, and the chromatin structure characteristic for PCs could not be organized. Given that H3.3 couples with the upregulation of gene expression, exogenous expression of H3.3 in B cells may regulate the expression of factors that suppress PC differentiation and maintain B cell identity. For example, overexpression of Bach2 or Bcl6 in B cells inhibits PC differentiation[16,19]. The levels of expression of these genes were downregulated during PC differentiation. This study showed that *Prdm1* and *Xbp1* were downregulated when H3.3 was expressed exogenously in B cells, even after PC differentiation was induced. H3.3 may

regulate such genes to suppress PC differentiation as a safeguard. However, we did not observe a correlation between H3.3 expression and DNA accessibility at these loci. Another possible scenario is that dynamic changes in H3.3 turnover affect PC differentiation. A decrease in newly synthesized H3.3 during PC differentiation would reduce H3.3 turnover efficiency. Thus, the dynamic changes in H3.3 turnover may also explain the effect of PC differentiation upon exogenous H3.3 expression. Further study is needed to determine why a large amount of H3.3 would prevent PC differentiation.

Our findings shed light on the role of the H3.3 histone variant in the process of PC differentiation. However, they do not rule out other potential models of PC differentiation, like those involving transcription factor networks and epigenetic regulation. Our study emphasizes the significance of additional layers of epigenetic remodeling, such as H3.3 distribution and chromatin accessibility, in understanding the PC differentiation process.

## Methods
### Mice
Mice were bred and maintained under the specific pathogen-free animal facility at Kyushu University in accordance with institutional guidelines under the following conditions: ambient temperature of 22 °C, 60% humidity, 12-h-dark and 12-h-light cycle, and free access to water and rodent chow (DC-8, CLEA Japan) and female mice were used at 8–12 weeks. All studies and procedures were approved by the Animal Experiment Committee of Kyushu University (Approval No: A23-090, A23-091, A23-103, A23-122, and A23-029). All animal experiments were conducted in accordance with the ARRIVE guidelines and the ethical guidelines of Kyushu University. C57BL/6 mice were purchased

from CLEA Japan. *Prdm1*[gfp/+] mice[59] and μMT mice[60] have been described previously.

## Antibodies

For flow cytometry or cell sorting, single-cell suspensions prepared from spleen, bone marrow, or cultured B cells were stained with the following fluorochrome-conjugated antibodies purchased from BioLegend, BD Bioscience, eBioscience or Invitrogen: Alexa647-conjugated anti-GFP (polyclonal); allophycocyanin (APC)-conjugated anti-CD19 (6D5), anti-CD138 (281-2) and anti-IgG1 (RMG1-1); Alexa647-conjugated and anti-IRF4 (IRF4.3E4); Pacific Blue (PB)-conjugated anti-B220 (RA3-6B2); Brilliant Violet (BV421)-conjugated anti-CD138 (281-2); Phycoerythrin (PE)-conjugated anti-TACI (8F10), anti-CD138 (281-2), and anti-Blimp-1 (5E7); Phycoerythrin-Cyanine7 (PE-Cy7)-conjugated anti-B220 (RA3-6B2) were used. For ChIL-seq, rat anti-H3.3 (4H2D7) homemade[61] and rat anti-GFP (1A5) purchased from Bio Academia Co. Ltd. were used. For western blotting, anti-H3.3 (6C4A3) and anti-H3.1/H3.2 (6G3C7) antibodies were produced in-house.

## Flow cytometry

Tissues were disrupted by passing through a nylon mesh (Kyoshin Ricoh). After red blood cell lysis with ammonium chloride potassium buffer, cells were incubated with an anti-CD16/CD32 (2.4G2; BD Pharmingen) to reduce nonspecific labeling of the cells before staining. Single cells were stained with fluorophore-labeled antibodies. Dead cells were excluded using propidium iodide (PI) (Nakalai), Zombie Aqua (Biolegend) or Zombie NIR (Biolegend). Intracellular staining was performed after fixation and permeabilization with the Foxp3/transcription factor staining buffer set (eBioscience). Data were acquired on a Cytoflex (Beckman Coulter) and analyzed with FlowJo software (Tree Star).

## Isolation and sorting of B cells

For B cell isolation, splenic B cells were purified by the negative selection of CD43$^+$ cells with anti-CD43 magnetic beads (Miltenyi Biotec). The purified B cell population was >95% positive for CD19 staining. Cell sorting was performed on FACSMelody (BD Biosciences) to isolate the following populations: activated B cells (CD138$^{low}$B220$^{hi}$), plasma cells (CD138$^{hi}$B220$^{low}$), and retrovirus-infected cells (GFP$^+$ or mCherry$^+$).

## Culture

For B cell stimulation assays, purified B cells (1 × 10$^6$ cells/ml) were cultured in RPMI 1640 medium supplemented with 10% (vol/vol) FCS, β-mercaptoethanol, L-glutamine, HEPES, NEAA and antibiotics. For plasma cell differentiation, B cells were stimulated with 5 μg/ml of lipopolysaccharide (LPS) (Sigma-Aldrich) or 1 μg/ml anti-CD40 antibody, 0.4 μg/ml IL-2, 5 ng/ml IL-4, and 2 ng/ml IL-5 for indicated days. For cell division assay, splenic B cells were labeled with Cell trace violet (CTV) (Thermo Fisher) followed by stimulation with LPS for 2 days. For cell apoptosis assays, cells were stained with APC Annexin V according to the manufacturer's protocol (BD Biosciences).

## Quantitative RT-PCR analysis

RNA was isolated and purified using the RNeasy kit (Qiagen) from negatively selected CD43$^-$ B cells, B220$^{hi}$CD138$^{low}$ activated B cells, and B220$^{low}$CD138$^{hi}$ plasma cells. cDNA was generated using the ReverTra Ace qPCR RT Master Mix (TOYOBO). Real-time PCR was performed on a LightCycler 96 (Roche) using Thunderbird SYBR qPCR mix (TOYOBO). The expression of target genes was normalized with 18S rRNA. The following primer pairs were used: *18S rRNA*, sense 5′-ATGGCCGTTCTTAGTTGGTG-3′ and antisense 5′-CGGACATCTAAGGGCATCAC-3′: *H3f3a*, sense 5′-ACTGGAGGGGTGAAGAAACC-3′ and antisense 5′-ACCAATAGCTGCACTCTGGAAG-3′: *H3f3b*, sense 5′-CAGGATTTCAAAACCGACTTGAG-3′ and antisense 5′-GTATCTTCAAACAA

CCCCACCAG-3′: *H3c1/2*, sense 5′-CTAAGCAGACCGCTCGCAAGTC-3′ and antisense 5′-CTTGAAGTCCTGCGCGATCTC-3′: *Prdm1*, sense 5′-CTATTAAGCCTATCCCTGCCAA-3′ and antisense 5′-GCTTTCCGTTTGTGTGAGATTTATC-3′: *Irf4*, sense 5′-ACAGCTCATGTGGAACCTCTG-3′ and antisense 5′-TCAGGTAACTCGTAGCCCCT-3′: *Irf5*, sense 5′-ATGTTGCCTTTGACGGACCTAG-3′ and antisense 5′-CAGGGCCAAAGAGTTCCACT-3′: *Pax5*, sense 5′-CAACAGGATCATTCGGACAA-3′ and antisense 5′-AGGATGCCACTGATGGAGTA-3′: *Bach2*, sense 5′-CATCTCTTCCTCTGCCCAGT-3′ and antisense 5′-CAGACATGCCGTTCAAACCATA-3′: *Irf8*, sense 5′-TGCAGGATGTGTGACCGGAA-3′ and antisense 5′-TAATCCTGCTTGCCGGCATG-3′: *Spi1*, sense 5′-GTCACCAGGTTTCCTACATGC-3′ and antisense 5′-CCAAGCCATCAGCTTCTCCA-3′.

## Plasmid, retrovirus production, and infection

The cDNA construction of N-terminally GFP-tagged histones H3.1 and H3.3 have been described previously[46]. GFP-H3.1 and GFP-H3.3 fragments were cloned into the pMY retroviral vector. mCherry-H3.1 and mCherry-H3.3 in pMY vector were similarly constructed. The point mutants of unique residues in H3.3 to residues in H3.1 (S31A or A87S or I89V or G90M or A87S, I89V, G90M) were generated by PCR. Nucleotide sequences of these constructs were verified by sequencing. To generate retroviral Blimp-1 expression vectors, cDNA corresponding to Blimp1 and IRF4 was obtained from mouse splenocytes by PCR amplification and then cloned into the pMY-IRES-GFP retroviral vector. An IRF4 retroviral expression vector was described previously[4]. For H3.3 knockdown, we designed the targeting sequence: 5′-CGAGAAATTGCTCAGGACTTC for H3f3a and 5′-CCAGAGATTGGTGAGGGAGAT for H3f3b. These miRNA gene double-strands were ligated with Block-iT™ Pol II GFP-miR RNAi expression vector (Invitrogen). Then, the resulting GFP-miR inserts were subcloned into a pMYs vector in tandem. The resulting vector or control vectors were transfected into the packaging cell line PLAT-E with FuGENE HD (Roche Diagnostics). On the following day of transfection, the medium was changed and cultured for 48 h. The virus supernatant was concentrated overnight using a Retro-X™ concentrator (Takara) according to the manufacturer's protocol. For infection of B cells, the splenic B cells were purified and stimulated with 5 μg/ml of LPS for 20 h. Cells underwent "spin infection" for 2 h at 25 °C (800 g) in concentrated virus supernatant containing polybrene (6 μg/ml). After spin infection, cells were washed with culture medium and cultured in addition 2 days with 5 μg/ml LPS for in vitro culture PC differentiation assay. For class switch assay, stimulated splenic B cells with 20 ng/ml of IL-4 (R&D) and 2 μg/ml of anti-CD40 (BD Biosciences) were infected as above and cultured with the same concentration of IL-4 and anti-CD40 for additional 3 days. For in vivo transfer assay, infected cells were cultured with 5 μg/ml LPS for additional 2 h and washed with PBS containing 3% FCS. The 2 × 10$^6$ infected B cells per head were transferred intravenously (i.v.) into μMT mice. The next day, the mice were injected intraperitoneally with 10 μg of LPS and were analyzed on day 3.

## Hydroxyapatite immunoprecipitation and Western blotting

Nucleosome levels were analyzed using samples purified by hydroxyapatite after the nuclear soluble fractions were removed, as described previously. In brief, cells were lysed in ice-cold physiological buffer (PB; 100 mM CH$_3$COOK, 30 mM KCl, 10 mM Na$_2$HPO$_4$, 1 mM DTT, 1 mM MgCl$_2$, 1 mM ATP, 0.1% Triton X-100 and protease inhibitor cocktail; Nacalai Tesque Inc.) and then centrifuged at 1000 × g for 5 min. The pellets were re-suspended in IP buffer (5 mM PIPES, 200 mM KCl, 1 mM CaCl$_2$, 1.5 mM MgCl$_2$, 5% sucrose, 0.5% NP-40, and protease inhibitor cocktail; Nacalai Tesque Inc.) and sonicated for 1 s three times and digested to lengths corresponding to mono-, di- and tri-nucleosomes with micrococcal nuclease (New England Biolabs) and Ribonuclease A (0.1 μg/ml) at 37 °C for 30 min. To stop the digestion reaction, ethylenediaminetetraacetic acid (EDTA) was added to the samples at a final concentration of 5 mM. 5 M NaCl was then added to achieve a

concentration of 0.5 M, and the samples were transferred to a spin column Mobicol "Classic" (MoBiTec GmbH), including hydroxyapatite resin (20 mg; Nacalai Tesque Inc.) pre-washed with hydroxyapatite buffer 1 (HAPB1; 5 mM NaPO₄, 600 mM KCl and 1 mM EDTA). The chromatin/hydroxyapatite mixture was incubated on a rotator for 30 min at 4 °C and washed twice with HAP1B and hydroxyapatite buffer 2 (HAPB2; 5 mM NaPO₄, 100 mM KCl, and 1 mM EDTA). The purified nucleosomes were eluted with hydroxyapatite (HAP) elution buffer (500 mM NaPO₄, 100 mM KCl, and 1 mM EDTA) and boiled in 5x SDS sample buffer. For H3.3 and H3.1/H3.2 expression, cells were lysed with PBS containing 0.5% Triton X-100 (v/v), protease inhibitor cocktail (Nakarai Tesque Inc.), and 1 mM EDTA and then centrifuged at $1000 \times g$ for 5 min. The pellets were re-suspended in 0.2 N HCL and neutralized with 2 M Tris-HCL (pH8.0) and 1 N NaOH. Samples were resolved on SDS/PAGE, transferred to a polyvinyldifluoride membrane (Millipore), and incubated with anti-H3.3 (6C4A3) or anti-H3.1/H3.2 (6G3C7) antibodies (homemade).

### Enzyme-linked immunosorbent assay (ELISA)
The concentration of total IgM in the culture supernatant was measured by ELISA with a plate coated with 0.5 µg/ml of goat anti-mouse IgM antibody (Southern Biotech). Plates were washed with PBS-0.05% Tween and incubated with diluted supernatant overnight at 4 °C. Plates were washed and incubated with HRP-conjugated goat anti-mouse IgM (1:3000) (Southern Biotech) for 3 h at room temperature. Plates were washed and developed with SureBlue TMB (KPL) until a blue color was visible. The reaction was stopped with 1 N HCl, and the absorbance was read at 450 nm immediately. The standard used to measure total IgM concentration was a monoclonal mouse IgM (Southern Biotech). For the quantitative measurement of IL-10, Purified B cells were cultured with 10 µg/ml of LPS (Sigma-Aldrich). The next day, cells were infected with retrovirus by spin infection protocol. Forty-eight hours after the start of the pre-stimulation, cells were additionally stimulated with 10 µg/ml of anti-IgM F(ab)′₂ (Jackson Immunoresearch). IL-10 in the culture medium was detected by ELISA assay according to the manufacturer's protocol (Biolegend).

### Fluorescence microscope
Cells stained with Hoechst (Nakalai) were allowed to settle on a glass bottom dish coated with Cell tak (Corning). All images were taken using the Keyence BZ-X700 All-In-One Fluorescence Microscope (Keyence Co.) and analyzed using BZ-X Analyzer version 1.3.1.1 (Keyence Co.).

### Chromatin integration labeling (ChIL)
ChIL-seq for B cells was performed slightly modified described previously[62]. Cells were biotinylated for 30 min at 4 °C with 1 mg/ml sulfo-NHS-SS-biotin in PBS and plated on streptavidin-treated 96-well plates (Thermo Fisher Scientific). After being fixed with 1% paraformaldehyde (Electron Microscopy Sciences) in PBS for 5 min at room temperature, fixation was stopped with 1 M Glycine/HCl. Cells were permeabilized with 1% Triton X-100 in PBS for 20 min and blocked with Blocking One-P (Nacalai Tesque) for 20 min at room temperature. Cells were incubated with 2 µg/ml rat anti-H3.3 or anti-GFP antibody in 0.1× Blocking One-P/PBS for 6 h at 4 °C and washed three times with PBS for 5 min each time, then incubated with 2 µg/ml of the ChIL probe (secondary antibody) at 4 °C overnight. After washing three times with PBS for 20 min each at 4 °C, PBS was removed from the wells, 50 µl Tn5 transposase solution (0.1 µl of 0.76 mg/ml Tn5 transposase) was added, and cells were incubated for 10 min at room temperature to allow Tn5 to bind to the mosaic end on ChIL DNA. 0.1 µl of Tn5-MEDS-B oligo (10 µM)[63] was then added at 50 µl per well in 1× dialysis buffer (50 mM HEPES-KOH, pH 7.2, 0.1 M NaCl, 0.1 mM EDTA, 1 mM dithiothreitol, 0.1% Triton X-100 and 10% glycerol) and cells were further incubated for 60 min at room temperature. After washing three times with PBS, the Tn5-mediated integration reaction was initiated by adding 50 µl of 1×

TAPS-DMF buffer (10 mM TAPS-NaOH, pH 8.5, 5 mM MgCl₂ and 10% *N,N*-dimethylformamide) for 1 h at 37 °C. In total, 100 µl of 50 mM EDTA were added to remove DNA-bound Tn5 and incubated for 30 min at 50 °C and washed three times with PBS without intervals. A sealing reaction was performed in 100 µl of reaction mixture (200 U T4 DNA ligase, NEB, and 1.5 U T4 DNA polymerase, NEB, in 1× T4 DNA ligase reaction buffer) for 30 min at room temperature. After discarding the solution, 100 µl of 50 mM EDTA was added, incubated for 30 min at 50 °C and washed three times with PBS to remove DNA-bound enzymes. In situ transcription was then carried out using Thermo T7 RNA polymerase (1000 U per well; Toyobo) or Ambion T7 RNA Polymerase (200 U per well; Thermo Fisher Scientific) for 3 h at 37 °C. The cells were treated with DNase I (2.5 U per well; TaKaRa) for 30 min at 37 °C, and the supernatant was then collected for RNA purification using AxyPrep Magnetic Bead (Axygen). Purified RNA was Fragmentated with 10× Fragmentation Buffer for 3 min at 94 °C and stopped with a 10× Fragmentation stop solution. After RNA purification, cDNA synthesis and amplification were performed using a SMART-Scribe Reverse Transcriptase (Clontech/Takara) following the manufacturer's protocol. The amplified DNA by PCR was purified using AMPure XP beads (Beckman Coulter), excluding fragments of <150 bp from the ChIL-seq library.

### Assay for transposase-accessible chromatin sequencing (ATAC-seq)
ATAC-seq was performed following published protocols[64]. Briefly, 50,000 cells were pelleted and washed in cold 1× PBS, centrifuged for 5 min at $500 \times g$ at 4 °C 2 times, and permeabilized with cold ATAC-Resuspension Buffer (RSB) (10 mM Tris-HCl, pH 7.4, 10 mM NaCl, 3 mM MgCl₂, 1% (vol/vol) containing NP40, Tween20 and 0.5% (vol/vol) containing Digitonin) for 3 min on ice. Cells were added ATAC-RSB and centrifuged for 10 min at $500 \times g$ at 4 °C. Tagmentation was performed on cell pellets by adding transposition mix (10 mM Tris-HCl, pH7.6, 5 mM MgCl₂,10% Dimethyl Formamide) containing 100 nM Tn5, 1% (vol/vol) containing Tween20 and 0.5% (vol/vol) containing Digitonin. After incubation for 30 min at 37 °C, reactions were purified using a DNA clean & concentrator kit (ZYMO Research) and eluted in Elution Buffer. Reactions were amplified by PCR using KAPA HiFi HS ReadyMix (KAPA BIOSYSTEMS) with Ad primer. After additional PCR, amplified DNA was purified using AMPure XP beads.

### Data analysis
ChIL-seq and ATAC-seq libraries were sequenced on an Illumina NovaSeq6000 as paired-end 50-base reads. After sequencing, reads were cleaned by trim_galore (version0.6.10) and mapped to mm10 with Bowtie2 (version 2.3.1). The read summary is shown in Supplementary Table 1. Replicate samples were merged and peaks were identified using MACS2 (ChIL: -q 0.01 --nomodel −nolambda, ATAC: -q 0.01). The genomic region profile of the peak was obtained by ChIPseeker. All ATAC samples were merged, and identified peaks by MACS2 (-q 0.01) were used as accessible chromatin regions. ATAC-seq and H3.3 ChIL-seq reads were counted on 1 Kbp or 2 Kbp flanking of accessible chromatin region center, respectively and 2Kbp flanking was annotated by overlapping with gene body for ChIL-seq analysis. Counts were used for PCA plot, and violin plot. We evaluated the difference of H3.3 enrichment using the linear mixed-effects model (lmerTest R package) to accommodate the unknown variation of batch effects. $p$ values were calculated from $t$-test (with Satterthwaite's approximation) using lmerTest (R package). Differential chromatin accessibility regions (DARs) for ATAC-seq and differential enrichment regions (DERs) for H3.3 ChIL-seq were obtained using DESeq2 at padj <0.1 as the threshold or absolute value of log2FoldChange >1, respectively. ATAC-seq signal on 3Kbp flanking of plasma cells DARs center is visualized using deeptools (version 3.5.1). Motif enrichment analysis was performed using 100 bp flanking of peak summit by MEME-ChIP (version 5.4.1). Bigwig files were generated to visualize

ChIL-seq and ATAC-seq using deeptools with the following options "--smoothLength 1000 --binSize 100 --normalizeUsing CPM" or "--binSize 100 --centerReads --normalizeUsing CPM" respectively. GO Enrichment Analysis is performed using "enrichGO" function of the R package "clusterProfiler". The correlation of ATAC-seq is visualized by scatter plot using read counts in 4 Kbp bins over the genome.

### Statistical analysis

We performed statistical evaluation using Prism software (GraphPad, La Jolla, CA). A two-tailed, unpaired $t$-test with Welch's correction was applied for the statistical comparison of two groups. Analysis of variance (ANOVA) was applied for statistical comparison between multiple groups. A $p$ value of less than 0.05 was considered statistically significant. The enrichment of H3.3 ChIL-seq peaks in promoter regions was analyzed using chi-square tests.

### Reporting summary

Further information on research design is available in the Nature Portfolio Reporting Summary linked to this article.

## Data availability

The ChIL-seq and ATAC-seq reported in this paper are available under Gene Expression Omnibus accession number GSE230496. All other data are available in the article and Supplementary Information or from the corresponding authors upon reasonable request. Source data are provided with this paper.

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

## Acknowledgements

We thank Kato, M. for ChIL-seq support; Tanaka, M. and Kageyama, K. for Technical Support in the Medical Institute of Bioregulation, Kyushu University for the animal facility: Furuno, N. for secretarial help. This work was partially supported by JSPS KAKENHI (JP20H05368 and JP21H05292 to A.H.; JP23H00372 and JP24H02323 to Y.O.: JP18H02626, JP19K22537, and JP21H02753 to Y.B.), AMED BINDS (JP22ama121017j0001 to Y.O.), JST PRESTO (JPMJPR19K7 to A.H.), Agency for Medical Research and Development (AMED) (JP19ek0410044, JP19gm6110004, JP23gm1810008 to Y.B.), a research grant from the GlaxoSmithKline (to Y.S.), the Astellas Foundation for Research on Metabolic Disorders and the Uehara Memorial Foundation (to Y.B.). This was also supported in part by the MEXT Promotion of Development of a Joint Usage / Research System Project :Pan-Omics DDRIC, MRCI for High Depth Omics, CURE:JPMXP1323015486 for MIB and RIIT in Kyushu University.

## Author contributions

Y.S. performed most of the experiments and wrote parts of the manuscript with assistance from M.U., R.H., D.M., T.N., and A.H. A.B. constructed all plasmids. A.H. conducted all the omics analyses with assistance from K.T. and wrote parts of the manuscript. S.L.N. provided advice and the important mouse strain. Y.O. acquired the funding support and designed the study. Y.B. designed and supervised the study and wrote most of the manuscript. All authors reviewed and commented upon the manuscript.

## Competing interests

A.H. and Y.O. have filed patent applications related to ChIL-seq (Application number: US11414680B2, JP6943376B2). The remaining authors declare no competing interests.
