## [Peer Review File · Nature Communications]

Plasma cell differentiation is regulated by the expression of histone variant H3.3Reviewers' Comments:

Reviewer #1:

Remarks to the Author:

Yuichi Saito et al. present a study investigating the role of the histone variant H3.3 during the differentiation of B cells into plasma cells. They demonstrate that H3.3 expression is downregulated during this process and that the overexpression of exogenous H3.3 inhibits plasma cell differentiation. This is characterized by a repression of the upregulation of plasma cell-associated genes (Irf4, Prdm1, and Xbp1) and a maintenance of the expression of B cell-associated genes (Pax5, Bach2, and Bcl6). The authors suggest this may be related to the maintenance of chromatin accessibility. Although this study presents intriguing insights into the role of histone variants in cell differentiation, several aspects require clarification. Specific comments and suggestions include:

1. The authors observed the downregulation of H3.3 during plasma cell differentiation, but the data does not conclusively demonstrate that this downregulation drives plasma cell differentiation. Experiments designed to knockdown H3.3 in B cells, to test if this induces plasma cell differentiation, may provide more insights on this question.
2. A more definitive experiment to identify the chaperone system (HIRA or DAXX/ATRX) involved in plasma cell differentiation could involve the knockdown or knockout of HIRA or DAXX/ATRX in B cells. This could shed more light on the role of specific chaperones in plasma cell differentiation.
3. In Figure 6, the use of GFP-H3.3 fusion proteins raises concerns due to the potential influence of the GFP tag on the deposition and function of H3.3. A mouse model using HA-tagged H3.3 might be a more suitable approach, especially considering the reported viability of H3.3B-HA mice, compared to the presumed lethality of GFP-H3.3 mice.
4. The statement on page 8, lines 10-12, "These data suggest that the restricted expression of H3.3 in PCs regulates cis-regulatory factors involved in gene expression during PC differentiation," is overreaching. The downregulation of H3.3 does not directly imply its role in specific gene regulation during plasma cell differentiation, considering the diverse changes in gene expression during this process.
5. The assertion on page 10, lines 9-10, that "These findings suggest that the inhibition of PC differentiation by H3.3 depends on specific chaperones," is not fully substantiated by the mutation of the H3.3 AIG motif alone. Given that H3.3 is known to associate with two chaperone systems (HIRA and ATRX/DAXX) that deposit it at different genome regions, it would be more conclusive to demonstrate the role of these systems by knockdown or knockout the HIRA or DAXX in plasma cell differentiation.

Minor comments:

In Figure 1d, the Western blot (WB) for H3.3 and H3.1/2 is missing.

Page 6, line 24, contains a typo: "naivee" should be "naive."

Page 7, line 5, has a typographical error: "biding profile" should be corrected to "binding profile."

Page 27, line 9, needs to provide more information regarding the antibodies specific for H3.1/H3.2. Are these pan-H3 antibodies?

Reviewer #2:

Remarks to the Author:

The manuscript by Saito et al. entitled "Plasma cell differentiation is regulated by the expression of histone variant H3.3" describes the role of H3.3 in preventing PC differentiation. This is a novel and under studied area of PC differentiation as well as a well-conceived study. Appropriate experiments were performed and results are adequately displayed in figures. While the exact mechanism is elusive, experiments have eliminated several obvious possibilities. Furthermore, the manuscript reveals new features of H3.3 that have not been previously described. Despite these many strengths, the text is often confusing and would benefit from a thorough editing. I would like to see this manuscript published with such an edit.

Major points:

1. Text. While there may be many more instances, these stood out:

- pg7, line 23: instead of "between the B" might it be "between the "activated" B?
- pg7, line 24-5: instead of "PCs in naïve", might it be "PCs "compared" to naïve"?
- pg9 line 6: it is unclear what "external" means
- pg13 line 13: might sentence beginning with "This" be written as "This might seem to contrast with the known roles of H3.3 in gene activation, a wide-spread process during PC differentiation."
- pg15 lines 11-3: unclear language of clause at end of sentence in line 11 and succeeding sentence beginning on line 12
- pg15 line 20: word "some" should be replaced with specific gene names

2. This study implicates the action of H3.3 chaperones; however, little discussion about the role of these chaperones in B cells and PC. Experiments to address the role of the chaperones might include measuring the expression levels of Hira, Daxx, Atrx during PC differentiation? What happens to PC differentiation if these chaperones ±H3.3 are transduced into activated B cells?

Reviewer #3:

Remarks to the Author:

In this study the authors examine H3.3 use in B cells and in vitro activated plasmablasts (PB) to show that H3.3 use is markedly reduced in committed PB, but its remaining deposition associates strongly with gene promoters, specifically at XBP1 and J-chain loci. In vivo, differentiation of LPS-activated B cells into PB was also diminished when H3.3 was overexpressed during a pre-transfer step. Overexpressing H3.3 in dividing B cells caused a reduction in PB formation or survival in culture, with mutations in the 87-90 AA region that interacts with chromatin modifiers able to revert the PB deficit to wildtype frequencies, showing that recruitment of chromatin modifiers was important for the impairment in PB production or survival caused by exogenous H3.3 overexpression. Overexpression of IRF4 or BLIMP1 could override the repression of differentiation, suggesting a complex mechanism by which H3.3 incorporation into nucleosomes restrains PB differentiation and/or survival. The study is interesting and to my knowledge, the first to probe H3.3 expression in the PC lineage.

Major points:

1) The study exclusively examines PB differentiation to a T-independent stimulus, LPS. To conclude that this is a core process required for differentiation of plasma cells generally, a T-dependent (anti-CD40) stimulus should be used to assess whether T cell help drives a similar H3.3 driven restraint of the differentiation of PB/PC in the exogenous H3.3 transduction setting.

Minor points:

2) Please amend Fig 1, 3,4,5 flow cytometry axis labels to state the fluorophore used for the antibodies.

3) For Fig 3f, please state how many days post immunization the analyses were performed in the legend. Second, how much LPS was given to the mice for the reactivation step. Third, what tissue was analysed? Peritoneal cavity? spleen? It is unclear from the legend, text and methods.

4) For Fig 3b and several figures, the gene KI could be affecting differentiation or survival. One might predict that there was no impact on B cell number relative to H3.1 KI, but the number of PC in the cultures reduced. Is this the case? Please show cell numbers.

5) For figure 4, please show total cell numbers of B cells and PB.

6) L24: naïve for naïvee

7) Discussion L11-13: H3.3 can't contrast with itself as is the argument of these two sentences. Please revise.

8) For Fig 2C/P6L7: Do the differences in promoter region associations differ statistically, e.g. via a chi-square test?

9) Results P9L13: While the overall division pattern looks similar, it looks like most cells have reached terminal division peak. I suggest the assay would be more sensitive to the point if the authors were to look before the mass proliferation has occurred. Also it isn't clear which analysis day this is from legend, methods or text so please state in figure legend to improve ease of reading. Alternative to

repeating the assays at an earlier timepoint, showing that B cell numbers are equivalent at this timepoint would be sufficient to address the contention that proliferation differs. That is, please show cell numbers or repeat the assays at an earlier timepoint.

10) Results P10L9: I think A87/I89/G90 should read A87SI89VG90M. Please also include a statement that S31A did not rescue the PC differentiation deficit, which is an important implication of the experiment, or state the limitation in the analysis that precludes such a conclusion.

11) Results P10L9: Presumably, mutating H3.1 to contain the DAXX/ATRX/HIRA motifs would prohibit PC differentiation as well? If the authors have performed such experiments to show a similar outcome it would make for helpful supplemental information.

12) For 6d, what does IRF4 look like?

13) P15L10: Please change "H3.3, IRF4, or BLIMP1" to "H3.3 with IRF4 or BLIMP1" or similar as meaning is unclear in current wording as exogenous expression of H3.3 diminishes PC production, but IRF4 and BLIMP1 overexpression with it reverses the suppression.

Response to reviewers

Our point-by-point responses to the issues raised by each reviewer are as follows. Sections with changes are indicated in blue in the revised manuscript. In the point-by-point below our responses are in blue:

Reviewer #1:

Yuichi Saito et al. present a study investigating the role of the histone variant H3.3 during the differentiation of B cells into plasma cells. They demonstrate that H3.3 expression is downregulated during this process and that the overexpression of exogenous H3.3 inhibits plasma cell differentiation. This is characterized by a repression of the upregulation of plasma cell-associated genes (*Irf4*, *Prdm1*, and *Xbp1*) and a maintenance of the expression of B cell-associated genes (*Pax5*, *Bach2*, and *Bcl6*). The authors suggest this may be related to the maintenance of chromatin accessibility. Although this study presents intriguing insights into the role of histone variants in cell differentiation, several aspects require clarification. Specific comments and suggestions include:

Reviewer #1 found our paper to be of significant interest but also made important points.

1. The authors observed the downregulation of H3.3 during plasma cell differentiation, but the data does not conclusively demonstrate that this downregulation drives plasma cell differentiation. Experiments designed to knockdown H3.3 in B cells, to test if this induces plasma cell differentiation, may provide more insights on this question.

We thank the reviewer for the comment. To address this question, we generated a retrovirus-based H3.3 knockdown vector (pMY-GFP-miR-H3f3a/H3f3b). Because B cells struggle to be transfected with siRNA, we utilized a retroviral transduction method, which provides high transduction efficiency and stable expression even after B cell proliferation. We confirmed the knockdown of *H3f3a/b* expression, and it enhanced plasma cell differentiation when stimulated with LPS (Figure for rebuttal [Figure R]1a, b) or anti-CD40 plus IL-2/IL-4/IL-5 (Figure R1c, d). The overexpression of H3.3 also suppresses the differentiation of antibody-secreting plasma cells induced by anti-CD40 and cytokines (see the response to Reviewer#3's comment 1) (Figure R2). We now include these data in the revised manuscript (Supplemental Figure 2e-j) (Page 9, Lines 15-19).

Figure R1

Figure R1. H3.3 knockdown enhanced PC differentiation.

For H3.3 knockdown, we designed the targeting sequence: 5'-CGAGAAATTGCTCAGGACTTC for H3f3a and 5'-CCAGAGATTGGTGAGGGAGAT for H3f3b. These miRNA gene double-strands were ligated with Block-iT™ Pol II GFP-miR RNAi expression vector (Invitrogen). Then, the resulting GFP-miR inserts were subcloned into a pMYs vector in tandem (mirH3.3). **a.** Representative flow cytometry plots of B cells retrovirally transduced with miR-H3.3 or control vector after stimulation with LPS for 2 days. Bars show the percentage of CD138⁺B220^{low} plasma cells (PCs) among GFP⁺ cells. **b.** Quantitative RT-PCR of H3f3a and H3f3b mRNA in purified GFP⁺ cells on day 3 after LPS stimulation was normalized to the expression of 18SrRNA **c.** Representative flow cytometry plots of B cells retrovirally transduced with miR-H3.3 or control vector after stimulation with anti-CD40 plus IL-2/IL-4/IL-5 for 5 days. Bars show the percentage of CD138⁺B220^{low} PCs among GFP⁺ cells. **d.** Quantitative RT-PCR of H3f3a and H3f3b mRNA in purified GFP⁺ B cells was normalized to the expression of 18SrRNA on day 3 after stimulation with anti-CD40 plus IL-2/IL-4/IL-5. Data are presented as mean ± SD of two independent experiments. *P < 0.05; **P < 0.01; ***P < 0.001; ****P < 0.0001; ns, not significant. The p values were obtained by a two-tailed unpaired t-test with Welch's correction.

Figure R2**Figure R2. Exogenous H3.3 expression inhibited PC differentiation when stimulated with anti-CD40 plus cytokines.**

a. Flow cytometry analysis of plasma cell differentiation in GFP (control), GFP-H3.3, or GFP-H3.1 expressing B cells on day 4 after stimulation with anti-CD40 plus IL-2/IL-4/IL-5. Bars show the percentage of CD138⁺B220^{low} plasma cells among GFP⁺ cells. **b.** The IgM antibody concentration in the culture supernatant of (a) was measured by ELISA. Data are representative of three independent experiments. Data are presented as mean ± SD. *P < 0.05; **P < 0.01; ***P < 0.001; ****P < 0.0001; ns, not significant. The p values were obtained by one-way ANOVA with Tukey's post hoc test.

2. A more definitive experiment to identify the chaperone system (HIRA or DAXX/ATRX) involved in plasma cell differentiation could involve the knockdown or knockout of HIRA or DAXX/ATRX in B cells. This could shed more light on the role of specific chaperones in plasma cell differentiation.

We first examined the expression levels of HIRA and DAXX/ATRX during PC differentiation and observed that they were downregulated in PCs compared to B cells (Figure R3).

Figure R3 Quantitative RT-PCR of mRNA encoding HIRA, DAXX, and ATRX in naive splenic B cells or CD138^{hi}B220^{low} plasma cells (PC) cultured with LPS for 4 days. Data are presented as mean ± SD of two independent experiments. *P < 0.05; **P < 0.01; ***P < 0.001; ****P < 0.0001; ns, not significant. The p values were obtained by a two-tailed unpaired t-test with Welch's correction.

To investigate the role of these chaperones, we constructed the knockdown vectors, pMY-GFP-miR-HIRA and pMY-GFP-miR-DAXX (designed for three different target sequences for each), and confirmed the reduced expression of Hira and Daxx in each vector-transduced B cells (Figure R4a). However, the knockdown of HIRA or DAXX did not seem to affect PC differentiation (Figure R4b).

Figure R4

Figure R4. Knockdown of HIRA and DAXX

a. B cells retrovirally transduced with miR-HIRA (mirHIRA) or miR-DAXX (mirDAXX) were stimulated with LPS for 3 days. For knockdown, we designed the targeting sequence: 5'-CTGGGTCAACCACAATGGGAA for HIRA and 5'-CCCGCTTGAAGAGGAAGTTGA for DAXX. These miRNA gene double-strands were ligated with Block-iT™ Pol II GFP-miR RNAi expression vector (Invitrogen). Then, the resulting GFP-miR inserts were subcloned into a pMYs vector. The expression of mRNA encoding HIRA or DAXX in purified GFP⁺ B cells was normalized to the expression of 18SrRNA.

b. Representative flow cytometry plots of B220 and CD138 expression on B cell transduced with control, mirHIRA or mirDAXX 3 days after LPS stimulation. Data are presented as mean ± SD of two independent experiments. *P < 0.05; **P < 0.01; ***P < 0.001; ****P < 0.0001; ns, not significant. The p values were obtained by a two-tailed unpaired t-test with Welch's correction (a) and by one-way ANOVA with Tukey's post hoc test (b).

There are several potential reasons for this observation. Firstly, it is possible that HIRA and DAXX may have complemented each other's roles or that other chaperones are involved in the process. Additionally, if H3.3 forms nucleosomes with different types of histones, it is possible that the deposition process may be carried out by chaperones independently of H3.3. Thirdly, the experiment was designed to further reduce the expression of these chaperones in the system where their expression is already low (Figure R3), making it difficult to assess the output accurately. Lastly, the knockdown efficiency may not have been sufficient for evaluation before differentiation. It should be noted that this system's limitation is that gene transfer to primary B cells by retrovirus requires pre-stimulation, but this stimulation initiates the PC differentiation program, thereby delaying the timing of knockdown. We believe that this strategy is technically a better choice at this stage. Although the H3.3 knockdown promoted PC differentiation described above (Figure R1), in the case of chaperones, the knockdown effect may be harder to see or may take more time since we have to observe an indirect effect. While we understand identifying the influence of specific chaperones on PC differentiation would be an interesting issue, presenting such data in a reliable and high-quality form would go beyond the intended scope and should be a future challenging issue. We were unable to determine the role of specific chaperones in PC differentiation, but we believe that our discovery related to H3.3 on PC differentiation is significant and unique at this point. We have mentioned this point in the revised version (Page 14, Lines 14-16).

3. In Figure 6, the use of GFP-H3.3 fusion proteins raises concerns due to the potential influence of the GFP tag on the deposition and function of H3.3. A mouse model using HA-tagged H3.3 might be a more suitable approach, especially considering the reported viability of H3.3B-HA mice compared to the presumed lethality of GFP-H3.3 mice.

We appreciate the comments. In Figure 6, we performed ChIL-seq using anti-H3.3 or anti-GFP antibodies in purified GFP-H3.3-transduced cells. So, anti-H3.3-ChIL evaluates the deposition of endogenous H3.3 and GFP-H3.3, but anti-GFP-ChIL can detect only the deposition of GFP-H3.3. Therefore, it is likely that the data comparing them are not identical. We noticed that the figure legend did not include the information, so we fixed it (Page 31, Line 1). However, as the reviewer pointed out, epitope tags may affect the function of H3.3. A study has shown that the expression of FLAG-FLAG-HA-H3.3 can prevent infertility in male mice deficient in H3f3b (Fontaine et al., 2022). However, another study reported that knock-in mice in which H3f3a and H3f3b genes were replaced by Flag-HA-tagged H3.3 had fewer offspring than the wild-type mice (Bachu et al. 2019). In this report, the number of pups from homozygous H3f3a-Flag-HA, H3f3b-Flag-HA, and H3f3a-HA/H3f3b-Flag-HA double knock-in mice was lower than expected when compared to wild-type mice. These data suggest that H3.3-Flag-HA fusion proteins may influence the function of H3.3. Therefore, concerns about the potential impact on the H3.3 function are unlikely to change even if GFP is substituted with HA or Flag tag. As we have observed that Flag-tagged H3.3 suppresses PC differentiation (Figure R5), we believe that switching to smaller tags would not alter the conclusions of this study. Embarrassingly, we could not find any report that the lethality of H3.3-deficient mice is not rescued by the expression of GFP-H3.3, as pointed out by Reviewer #1. Instead, several reports show the GFP-tagged H3 variants look normal functionally in embryo development, genome deposition, and chromatin binding (Santenard et al. 2010; Harada et al. 2015; Harada et al. 2017). We agree this is an important point, but the data is too large to repeat using H3.3-HA or a different tag instead of the GFP. We believe that our findings that H3.3 is downregulated during PC differentiation and enforced expression of GFP-H3.3 block the effect are informative and have sufficient novelty.

Figure R5

Figure R5. Splenic B cells were stimulated with LPS and retrovirally transduced with pMY-GFP control or pMY-Flag-H3.3-IRES-GFP vector. Plasma cell differentiation was assessed by flow cytometry analysis in GFP (control) or Flag-H3.3-IRES-GFP expressing B cells on day 4 after LPS stimulation. Bar graph indicates the percentage of CD138⁺B220^{low} plasma cells among GFP⁺ cells. Data are presented as mean \pm SD of two independent experiments. *P < 0.05; The p values were obtained by a two-tailed unpaired t-test with Welch's correction.

References

- Fontaine, E. et al. Dual role of histone variant H3.3B in spermatogenesis: positive regulation of piRNA transcription and implication in X-chromosome inactivation. *Nucleic Acids Res.* 50, 7350–7366 (2022).
- Bachu, M. et al. A versatile mouse model of epitope-tagged histone H3.3 to study epigenome dynamics. *J Biol Chem* 294, 1904–1914 (2019).
- Santenard, A. et al. Heterochromatin formation in the mouse embryo requires critical residues of the histone variant H3.3. *Nat. Cell Biol.* 12, 853–862 (2010).
- Harada, A. et al. Incorporation of histone H3.1 suppresses the lineage potential of skeletal muscle.

Nucleic Acids Research 43, 775–786 (2015).

Harada, A. et al. Histone H3.3 sub-variant H3mm7 is required for normal skeletal muscle regeneration. *Nat. Commun.* 9, 1400 (2018).

4. The statement on page 8, lines 10-12, “These data suggest that the restricted expression of H3.3 in PCs regulates cis-regulatory factors involved in gene expression during PC differentiation,” is overreaching. The downregulation of H3.3 does not directly imply its role in specific gene regulation during plasma cell differentiation, considering the diverse changes in gene expression during this process.

We agree with this remark and have modified the text (Page 7, Lines 17-18).

5. The assertion on page 10, lines 9-10, that "These findings suggest that the inhibition of PC differentiation by H3.3 depends on specific chaperones," is not fully substantiated by the mutation of the H3.3 AIG motif alone. Given that H3.3 is known to associate with two chaperone systems (HIRA and ATRX/DAXX) that deposit it at different genome regions, it would be more conclusive to demonstrate the role of these systems by knockdown or knockout the HIRA or DAXX in plasma cell differentiation.

See comment to your second major comment above. Because we did not reach a concrete conclusion, we have revised the manuscript to describe it more accurately (Page 10, Lines 15-17).

Minor comments:

In Figure 1d, the Western blot (WB) for H3.3 and H3.1/2 is missing.

According to the comment, we performed a WB analysis on H3.3 and H3.1/3.2. We stimulated B cells with LPS for 4 days, and sorted them into PCs and non-PCs (activated B cells; Division 6). However, we faced a shortage of cells in D0, which made WB analysis impossible. The WB analysis confirmed the lowered H3.3 expression in PCs (Figure R6). The revised manuscript now includes this as Figure 1e.

Figure R6

Figure R6. Western blotting analysis of LPS-induced PC and non-PC (activated B; D6) cells with antibodies specific for H3.1/H3.2 and H3.3.

Page 6, line 24, contains a typo: "naivee" should be "naive."

Thanks for pointing out the typo. We fixed this in the revised manuscript (Page 6, Line 24).

Page 7, line 5, has a typographical error: "biding profile" should be corrected to "binding profile."

Thanks. We fixed this in the revised manuscript (Page 7, Line 5).

Page 27, line 9, needs to provide more information regarding the antibodies specific for H3.1/H3.2. Are these pan-H3 antibodies?

The original manuscript lacked information on the antibody. This self-made monoclonal antibody is specific for H3.1 and H3.2 (6G3C7) rather than pan-H3. The revised manuscript now includes a statement about this in the legend and method section (Page 17, Line 21; Page 27, Lines 15-16).

Finally, we are grateful to Reviewer#1 for the critical comments and useful suggestions that have helped us to improve our manuscript considerably.

Reviewer #2:

The manuscript by Saito et al. entitled “Plasma cell differentiation is regulated by the expression of histone variant H3.3” describes the role of H3.3 in preventing PC differentiation. This is a novel and under studied area of PC differentiation as well as a well-conceived study. Appropriate experiments were performed and results are adequately displayed in figures. While the exact mechanism is elusive, experiments have eliminated several obvious possibilities. Furthermore, the manuscript reveals new features of H3.3 that have not been previously described. Despite these many strengths, the text is often confusing and would benefit from a thorough editing. I would like to see this manuscript published with such an edit.

Reviewer #2 praised the study as a whole but also raised important questions.

Major points:

1. Text. While there may be many more instances, these stood out:

- pg7, line 23: instead of “between the B” might it be “between the “activated” B?

Thanks. This should be “naive B cells”. We have modified the text (Page 7, Line 24).

- pg7, line 24-5: instead of “PCs in naïve”, might it be “PCs “compared” to naïve”?

We have fixed it in the text of the revised manuscript accordingly (Page 8, Line 1).

- pg9 line 6: it is unclear what “external” means

This meant “enforced expression”. We have modified the text (Page 9, Line 7).

- pg13 line 13: might sentence beginning with “This” be written as “This might seem to contrast with the known roles of H3.3 in gene activation, a wide-spread process during PC differentiation.”

We appreciate the suggestion and have modified the text accordingly (Page 13, Lines 13-14).

- pg15 lines 11-3: unclear language of clause at end of sentence in line 11 and succeeding sentence beginning on line 12

We agree that this statement is difficult to understand and revised it to make it simple and clear (Page 15, Lines 12-14).

- pg15 line 20: word “some” should be replaced with specific gene names

As the reviewer pointed out, the revised manuscript now includes specific gene names (Page 15, Line 21).

2. This study implicates the action of H3.3 chaperones; however, little discussion about the role of these chaperones in B cells and PC. Experiments to address the role of the chaperones might include measuring the expression levels of Hira, Daxx, Atrx during PC differentiation? What happens to PC differentiation if these chaperones ±H3.3 are transduced into activated B cells?

Thank you for the valuable suggestion. As stated in our response to Reviewer #1's comments 2 and 5, we found that Hira, Daxx, and ATRX expression was downregulated in plasma cells (PCs) compared to B cells (Figure R3). According to Reviewer#2's suggestion, we transduced the pMYs-HIRA-IRES-mCherry or control vector into activated B cells in the presence or absence of GFP-H3.3. We observed that enforced H3.3 expression suppressed PC differentiation, but co-expression of HIRA and H3.3 restored it (Figure R7). However, HIRA expression alone did not

affect PC differentiation (Figure R7). We initially hypothesized that if we forcibly expressed HIRA, it would strengthen the H3.3-mediated inhibition of PC differentiation, but the results were the opposite. While we do not have a clear answer, it is possible that HIRA may actually contribute to the inhibition of PC differentiation by H3.3, but at the same time, HIRA may also positively regulate the expression of IRF4 or Blimp1 in an H3.3-dependent manner, which leads to the induction of PC differentiation. Indeed, HIRA has been reported to increase the accessibility of the *Irf4* gene and regulate its mRNA expression (Tamura et al., 2020). In addition, our data indicate that co-expression of H3.3 with IRF4 or Blimp1, even when H3.3 is enforced, can counteract the H3.3-mediated suppression of PC differentiation. We also attempted to co-transduce both DAXX and ATRX chaperones in B cells since ATRX and DAXX complex is necessary for H3.3 deposition at the genome (Goldberg et al. 2010; Drane et al. 2010; Lewis et al. 2010). However, the size of ATRX (7431 bp) prevented successful transduction. In general, transferring genes into primary B cells is a challenging task, and currently, there are two successful methods available: retroviruses (or lentiviruses) and electroporation. However, electroporation has a very low transduction efficiency, which makes it quite difficult to study the biology of transduced cells. On the other hand, viral transduction is more efficient, but it has a limitation. It can only package genes up to a certain size, which makes it ineffective for large genes. This limitation may be overcome in the future as technology advances and allows for the introduction of large genes into primary B cells. We could not identify the involvement of enforced expression of specific chaperones, but we believe that our new findings about H3.3 on PC differentiation are important and sufficiently novel at this stage. We have mentioned this point in the revised version (Page 14, Lines 14-16).

Figure R7

Figure R7 a. Flow cytometry analysis of LPS-induced CD138^{hi}B220^{low} plasma cell differentiation after enforced expression of HIRA (mCherry) and/or GFP-H3.3. Data are representative of two independent experiments. Data are presented as mean \pm SD. *P < 0.05; **P < 0.01; ***P < 0.001; ****P < 0.0001; ns, not significant. The p values were obtained by one-way ANOVA with Tukey's post hoc test.

References

- Tamura, T. et al. Inducible Deposition of the Histone Variant H3.3 in Interferon-stimulated Genes*. *J Biol Chem* 284, 12217–12225 (2009).
- Goldberg AD, Banaszynski LA, Noh KM, Lewis PW, Elsaesser SJ, Stadler S, Dewell S, Law M, Guo X, Li X, et al. 2010. Distinct factors control histone variant H3.3 localization at specific genomic regions. *Cell* 140: 678–691.
- Drane P, Ouararhni K, Depaux A, Shuaib M, Hamiche A. 2010. The death-associated protein DAXX is a novel histone chaperone involved in the replication-independent deposition of H3.3. *Genes Dev* 24: 1253–1265

Lewis PW, Elsaesser SJ, Noh KM, Stadler SC, Allis CD. 2010. Daxx is an H3.3-specific histone chaperone and cooperates with ATRX in replication-independent chromatin assembly at telomeres. *Proc Natl Acad Sci* 107: 14075–14080.

Finally, we would like to express our appreciation for the rigorous and constructive review of our manuscript.

Reviewer #3

In this study the authors examine H3.3 use in B cells and in vitro activated plasmablasts (PB) to show that H3.3 use is markedly reduced in committed PB, but its remaining deposition associates strongly with gene promoters, specifically at XBP1 and J-chain loci. In vivo, differentiation of LPS-activated B cells into PB was also diminished when H3.3 was overexpressed during a pre-transfer step. Overexpressing H3.3 in dividing B cells caused a reduction in PB formation or survival in culture, with mutations in the 87-90 AA region that interacts with chromatin modifiers able to revert the PB deficit to wildtype frequencies, showing that recruitment of chromatin modifiers was important for the impairment in PB production or survival caused by exogenous H3.3 overexpression. Overexpression of IRF4 or BLIMP1 could override the repression of differentiation, suggesting a complex mechanism by which H3.3 incorporation into nucleosomes restrains PB differentiation and/or survival. The study is interesting and to my knowledge, the first to probe H3.3 expression in the PC lineage.

This is a very positive review in terms of quality and impact.

Major points:

1) The study exclusively examines PB differentiation to a T-independent stimulus, LPS. To conclude that this is a core process required for differentiation of plasma cells generally, a T-dependent (anti-CD40) stimulus should be used to assess whether T cell help drives a similar H3.3 driven restraint of the differentiation of PB/PC in the exogenous H3.3 transduction setting.

We appreciate the suggestion. To address this question, we examined PC differentiation induced by anti-CD40 and cytokines (IL-2, IL-4, and IL-5) in control and GFP-H3.3/H3.1-transduced B cells. We found that the transduction of H3.3 significantly decreased the frequency of PC development and antibody production, whereas control and H3.1 expression had no effect on it (Figure R2). This was essentially the same result as for LPS-induced PCs. Therefore, we believe that this is a fundamental process required for the differentiation of PCs in general. These data have been added to the revised manuscript as Supplemental Figure 2e.

Figure R2

Figure R2. Exogenous H3.3 expression inhibited PC differentiation when stimulated with anti-CD40 plus cytokines.

a. Flow cytometry analysis of plasma cell differentiation in GFP (control), GFP-H3.3, or GFP-H3.1 expressing B cells on day 4 after stimulation with anti-CD40 plus IL-2/IL-4/IL-5. Bars show the percentage of CD138⁺B220^{low} plasma cells among GFP⁺ cells. **b.** The IgM antibody concentration in the culture supernatant of (a) was measured by ELISA. Data are representative of three independent experiments. Data are presented as mean \pm SD. *P < 0.05; **P < 0.01; ***P < 0.001; ****P < 0.0001; ns, not significant. The p values were obtained by one-way ANOVA with Tukey's post hoc test.

Minor points:

2) Please amend Fig 1, 3,4,5 flow cytometry axis labels to state the fluorophore used for the antibodies.

We have modified the figures accordingly.

3) For Fig 3f, please state how many days post immunization the analyses were performed in the legend. Second, how much LPS was given to the mice for the reactivation step. Third, what tissue was analysed? Peritoneal cavity? spleen? It is unclear from the legend, text and methods.

We agree with Reviewer#2's comments that the experimental conditions were not adequately explained. In this experiment, we adoptively transferred B cells transduced with GFP (control) or GFP-H3.3 to μ MT mice and then immunized with 10 μ g/mL LPS. After 3 days, we conducted a flow cytometry analysis of the spleen. We revised Figure 3f and text in the method and legend section (Page 21, Line 12; Page 28, Lines 14-16).

4) For Fig 3b and several figures, the gene KI could be affecting differentiation or survival. One might predict that there was no impact on B cell number relative to H3.1 KI, but the number of PC in the cultures reduced. Is this the case? Please show cell numbers.

According to the suggestion, the plasma cell numbers in Figure 3b were indicated (Figure R8), which gave the same results as looking at frequencies. This data is now included in Figure 3b of the revised manuscript.

Figure R8

Figure R8. The bars show plasma cell numbers from GFP (control), GFP-H3.3, or GFP-H3.1-expressing samples on day 4 after LPS stimulation in Figure 3b. Data are pooled from two independent experiments. Data are presented as mean \pm SD. *P < 0.05; **P < 0.01; ***P < 0.001; ****P < 0.0001; ns, not significant. The p values were obtained by one-way ANOVA with Tukey's post hoc test.

5) For figure 4, please show total cell numbers of B cells and PB.

According to the suggestion, the cell numbers of B cells and plasma cells have been shown in Figure R9. These results were essentially the same as when looking at frequencies (Figure 4b).

Figure R9

Figure R9. The bars show numbers of CD138^{hi}B220^{hi} activated B cells or CD138^{hi}B220^{low} plasma cells from GFP (control), GFP-H3.3, GFP-H3.1, GFP-H3.3 S31A, GFP-H3.3 A87S, GFP-H3.3 I89V, GFP-H3.3 G90M, or GFP-H3.3 A87S/I89V/G90M expressing samples on day 4 after LPS stimulation in Figure 4b. Data are pooled from two independent experiments. Data are presented as mean ± SD. *P < 0.05; **P < 0.01; ***P < 0.001; ****P < 0.0001; ns, not significant. The p values were obtained by one-way ANOVA with Tukey's post hoc test.

6) L24: naïve for naïvee.

Thanks for pointing out the typo. We fixed this in the revised version of the manuscript (Page 6, Line 24).

7) Discussion L11-13: H3.3 can't contrast with itself as is the argument of these two sentences. Please revise.

According to the reviewer's suggestion, we revised the sentence to make it clear (Page 13, Lines 12-15).

8) For Fig 2C/P6L7: Do the differences in promoter region associations differ statistically, e.g. via a chi-square test?

Using the chi-square test, we evaluated whether the H3.3 ChIL-seq peaks overlapping with the promoter were significantly enriched in plasma cells compared to naïve and activated B cells. The results indicated a significant enrichment compared to what would be expected by chance ($p=1.99214 \times 10^{-72}$ for Naïve, $p=4.2276 \times 10^{-51}$ for Activated B). We added the information, and the text in the Results, Methods, and Legend sections was revised (Page 7, Lines 6-7; Page 26, Lines 13-16; Page 28, Lines 5-8)

9) Results P9L13: While the overall division pattern looks similar, it looks like most cells have reached terminal division peak. I suggest the assay would be more sensitive to the point if the authors were to look before the mass proliferation has occurred. Also it isn't clear which analysis day this is from legend, methods or text so please state in figure legend to improve ease of reading. Alternative to repeating the assays at an earlier timepoint, showing that B cell numbers are equivalent at this timepoint would be sufficient to address the contention that proliferation differs. That is, please show cell numbers or repeat the assays at an earlier timepoint.

According to the reviewer's appropriate suggestion, we revised the manuscript to include the information in the legend and method section. We confirmed that H3.3 expression did not influence B cell proliferation at an earlier time point (LPS stimulation for 2 days) (Figure R10). Because we agree with the reviewer's point that most cells in the original paper appear to have reached peak terminal division, we have replaced it with the new data shown in Figure 3d.

Figure R10

Figure R10. Proliferation of GFP⁺ B cells transduced with GFP (control), GFP-H3.3, or GFP-H3.1 labeled with CTV and stimulated for 2 days with LPS. Percentages of proliferating B cells were assessed by CTV dilution.

10) Results P10L9: I think A87/I89/G90 should read A87SI89VG90M. Please also include a statement that S31A did not rescue the PC differentiation deficit, which is an important implication of the experiment, or state the limitation in the analysis that precludes such a conclusion.

We thank the reviewer for the comment. “A87S/I89V/G90M” is correct, so we have modified the text accordingly. In the revised manuscript, we also add the statement, “The S31A mutation did not affect the inhibitory effect of H3.3 on PC differentiation” (Page 10, Lines 15-16).

11) Results P10L9: Presumably, mutating H3.1 to contain the DAXX/ATR/HIRA motifs would prohibit PC differentiation as well? If the authors have performed such experiments to show a similar outcome it would make for helpful supplemental information.

We appreciate the suggestion, but we believe that the H3.1 mutant carrying the DAXX/ATR/HIRA motifs (H3.1 S87A/V89I/M90G) is identical to H3.3 S31A. Therefore, the mutant should be able to inhibit PC differentiation.

12) For 6d, what does IRF4 look like?

As suggested by the reviewer, we have shown the ATAC-seq data on the *Irf4* gene. We added this data in Figure 6d of the revised manuscript.

Figure R11. Genome browser images showing chromatin accessibility on the typical genes.

13) P15L10: Please change “H3.3, IRF4, or BLIMP1” to “H3.3 with IRF4 or BLIMP1” or similar as meaning is unclear in current wording as exogenous expression of H3.3 diminishes PC production, but IRF4 and BLIMP1 overexpression with it reverses the suppression.

According to your miner comment 2 and Reviewer#2’s comment, this sentence was revised. However, we have revised the manuscript to reflect the text you suggested (Page 14, Lines 18-20).

Again, we thank the reviewer for bringing important points to our attention and helping to improve our manuscript.

Reviewers' Comments:

Reviewer #1:

Remarks to the Author:

In this revised manuscript, the authors demonstrated that during plasma cell differentiation, H3.3 is downregulated. Blockade of H3.3 downregulation by enforced H3.3 overexpression impairs plasma cell differentiation. Exogenous overexpression of H3.3 inhibits the upregulation of plasma cell-associated genes such as *Irf4*, *Prdm1*, and *Xbp1*, and maintains the expression of B cell-associated genes, *Pax5*, *Bach2*, and *Bcl6*. The authors concluded that appropriate H3.3 expression drives plasma cell differentiation. Although the authors in this revision have addressed most of my concerns, I support the acceptance for publication after some revision.

Specific points: In page 4, line 3-4, please check the accuracy of the statement: "Although H3.3 expression did not affect B cell proliferation and apoptosis, it suppressed the expression of *Irf4*, *Prdm1*, and *Xbp1* and downregulated *Pax5*, *Bcl6*, and *Bach2*." According to the result, overexpression of H3.3 leads to the upregulation of *Pax5*, *Bcl6*, and *Bach2* (Fig. 5a).

Fig.1e is not cited in the text.

Provide the values in Fig. 2c.

Please revise the sentence: "These data suggest that during the differentiation of naive B cells into PCs, the deposition of H3.3 is somewhat correlated with gene expression at each stage of differentiation." on page 7. "Somewhat correlated" is too vague.

In page 8, line 18, and page 12, line 1, GFP-H3.3 does not replace but competes with endogenous H3.3; please revise.

Reviewer #2:

Remarks to the Author:

Revised manuscript is suitable for publication.

Reviewer #3:

Remarks to the Author:

Figure 1e needs to be referenced in the results section of the manuscript
y-axis labels in Supplementary figure 2h and 2j should be included.

Response to reviewers

Our point-by-point responses to the issues raised by each reviewer are as follows. In the revised manuscript, changes are indicated in blue. In the point-by-point below our responses are in blue:

Reviewer #1:

In this revised manuscript, the authors demonstrated that during plasma cell differentiation, H3.3 is downregulated. Blockade of H3.3 downregulation by enforced H3.3 overexpression impairs plasma cell differentiation. Exogenous overexpression of H3.3 inhibits the upregulation of plasma cell-associated genes such as Irf4, Prdm1, and Xbp1, and maintains the expression of B cell-associated genes, Pax5, Bach2, and Bcl6. The authors concluded that appropriate H3.3 expression drives plasma cell differentiation. Although the authors in this revision have addressed most of my concerns, I support the acceptance for publication after some revision.

Specific points: In page 4, line 3-4, please check the accuracy of the statement: "Although H3.3 expression did not affect B cell proliferation and apoptosis, it suppressed the expression of Irf4, Prdm1, and Xbp1 and downregulated Pax5, Bcl6, and Bach2." According to the result, overexpression of H3.3 leads to the upregulation of Pax5, Bcl6, and Bach2 (Fig. 5a).

We appreciate the comment. I wanted to say that Pax5, Bcl6, and Bach2 are downregulated during plasma cell differentiation, but that H3.3 expression inhibited this downregulation. We revised the sentence to make it clear (Page 5, Lines 2-3).

Fig. 1e is not cited in the text.

Thanks for pointing out the missing information. We fixed this in the revised version of the manuscript (Page 6, Line 19).

Provide the values in Fig. 2c.

According to the suggestion, the values were indicated in Fig. 2c. This data is now included in supplemental table 1 of the revised manuscript.

Please revise the sentence: "These data suggest that during the differentiation of naive B cells into PCs, the deposition of H3.3 is somewhat correlated with gene expression at each stage of differentiation." on page 7. "Somewhat correlated" is too vague.

According to the reviewer's suggestion, we revised the text (Page 7, Lines 18-19).

In page 8, line 18, and page 12, line 1, GFP-H3.3 does not replace but competes with endogenous H3.3; please revise.

We revised the text accordingly.

We appreciate Reviewer#1 for their critical feedback and valuable suggestions that significantly enhanced our manuscript.

Reviewer #3:

Figure 1e needs to be referenced in the results section of the manuscript.

Thanks for pointing out the missing information. We fixed this in the revised version of the manuscript (Page 6, Line 19).

y-axis labels in Supplementary figure 2h and 2j should be included.

Thanks for pointing out the missing information. We included the y-axis labels in Supplementary Figure 2h and 2j.

We express our gratitude for the thorough and beneficial review of our manuscript.